# MedMosaic: A Challenging Large Scale Benchmark of Diverse Medical Audio

**Harshit Rajgarhia** [* 1]  **Shuubham Ojha** [* 2 1]  **Asif Shaik** [* 1]  **Akhil Pothanapalli** [* 1]  **Rachuri Lokesh** [* 1]
**Abhishek Mukherji** [† 1]  **Prasanna Desikan** [† 1]

## Abstract

Medical audio data is difficult to collect due to privacy regulations and high annotation costs arising from domain expertise. Thus, existing benchmarks tend to underrepresent complex medical audio scenarios. To address this challenge, we present MedMosaic, a medical audio question–answering dataset designed to benchmark language and audio reasoning models under realistic clinical constraints. MedMosaic features a diverse range of medical audio types, including condition-related physiological sounds, carefully constructed synthetic voices to mimic speech with artifacts as well as real short and long length clinical conversations to model varying context lengths. The dataset also features a total of 46,701 question-answer pairs, spanning categories such as multiple-choice, sequential multi-turn, and open-ended question–answers, enabling systematic evaluation of multi-hop reasoning and answer generation capabilities. Benchmarking 13 audio and multimodal reasoning models reveals that reasoning remains challenging for all evaluated systems, with substantial performance variation across question types. In particular, even state-of-the-art model like Gemini-2.5-pro can only achieve 68.1% accuracy approximately. These findings underscore persistent limitations in medical reasoning and highlight the need for more robust, domain-specific multimodal reasoning models. A sample of benchmark data is available here:https://shorturl.at/Lyp33

## 1. Introduction

Reasoning over complex, structured inputs has become a central objective in evaluating modern language and multimodal models. Beyond surface-level recognition or factual recall, contemporary evaluation increasingly focuses on whether models can integrate evidence over time, maintain contextual state, perform abstraction, and draw inferences under uncertainty. Such capabilities are essential in real-world settings, where relevant information is often sparse, noisy, and temporally distributed.

In particular, reasoning over medical audio is challenging since existing benchmarks focus on short audio segments, single-turn questions, or environmental sounds, limiting evaluation of long-range temporal reasoning, multi-turn interactions, and subtle domain-specific cues. Such constraints make it difficult to systematically study reasoning ability of models over real-world clinical audio, where evidence may be sparse. Many clinically meaningful signals such as respiratory sounds, vocal strain, hesitations, and prosodic markers of discomfort are only conveyed through audio. Clinical decision-making crucially hinges on aligning spoken content with such medical acoustic markers, a feature missing from existing audio benchmarks.

Recent advances in large language models (LLMs), multimodal large language models (MLLMs), and large audio–language models (LALMs) have further intensified the focus on multimodal reasoning. Models trained jointly over text, audio, and vision increasingly exhibit emergent inference behaviors, including temporal abstraction, cross-modal grounding, and causal reasoning. This progress has motivated a shift in benchmarking away from low-level modality recognition toward the evaluation of higher-level reasoning abilities.

Building on this paradigm, benchmarks such as MMLU (Hendrycks et al., 2021) formalize multi-step, compositional reasoning across diverse subjects, motivating analogous efforts in the audio domain. Recent benchmarks including MMAU (Sakshi et al., 2025) and MMAU-Pro (Kumar et al., 2025), MDAR (Li et al., 2025b), MMAR (Ma et al., 2025), and AudioBench (Wang et al., 2025a) extend rigorous reasoning evaluation to speech, environmental sound, and music by probing temporal reasoning, causal inference, event tracking, hierarchical inference, and cross-modal consistency under increasingly complex acoustic conditions. In parallel, architectures such as Audio

---
[*]Equal contribution [†]Equal advising [1]Centific Global Solutions Inc. [2]University of Maryland, College Park, MD, USA. Correspondence to: Harshit Rajgarhia <harshit.rajgarhia@centific.com>.

*Proceedings of the 43$^{rd}$ International Conference on Machine Learning*, Seoul, South Korea. PMLR 306, 2026. Copyright 2026 by the author(s).

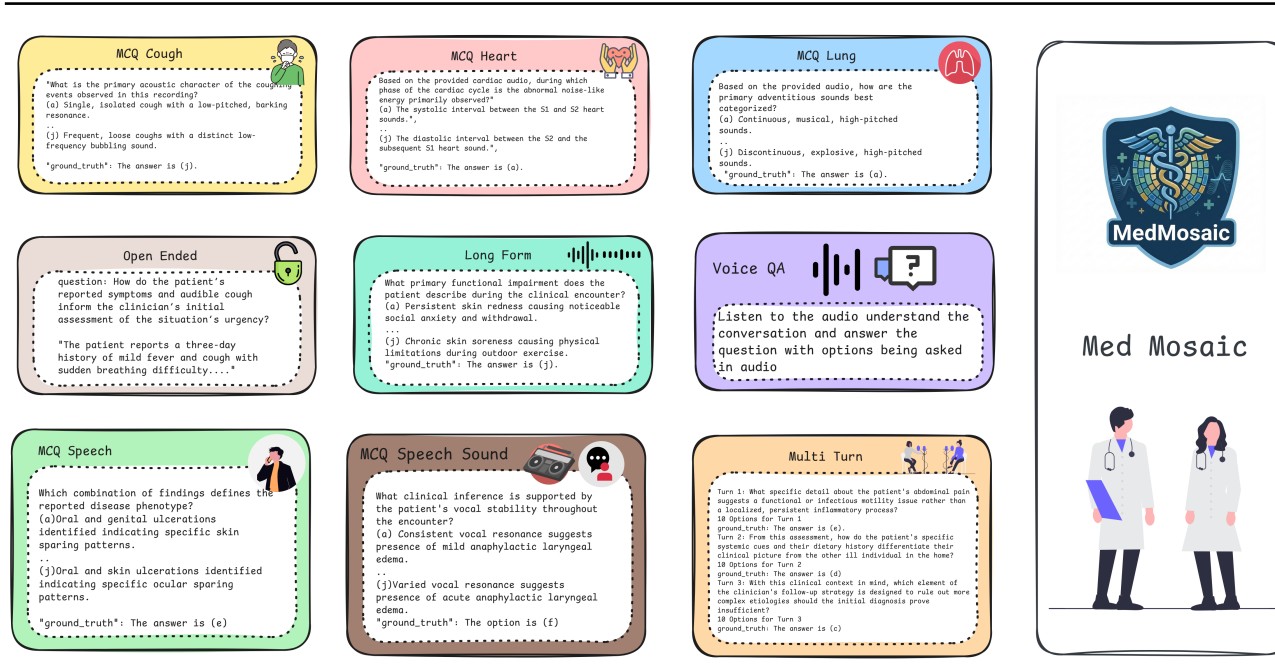

*Figure 1.* An overview of the structure of MedMosaic and its constituent QA pair categories.

Flamingo (Kong et al., 2024) adapt gated cross-attention and in-context learning to audio–text inputs, enabling few-shot reasoning over long, structured acoustic sequences. Despite these advances, most large audio–language models remain domain-specialized, with distinct model families for environmental sounds, speech, or music (e.g., Pengi (Deshmukh et al., 2023), LTU (Gong et al., 2024), and GAMA (Ghosh et al., 2024) for sound; SALM (Chen et al., 2023) and AudioPaLM (Rubenstein et al., 2023) for speech; LLark (Gardner et al., 2024) and MERT (LI et al., 2024) for music), and only a few models, such as Qwen-Audio (Chu et al., 2023) and Audio Flamingo, demonstrating more unified cross-domain audio understanding, leaving evaluation largely constrained to domain-specific benchmarks.

To address this gap, we introduce a medical audio question-answering benchmark designed to evaluate reasoning over diverse acoustic evidence under realistic, low-resource conditions. Our contribution is four-fold:

- **A medical audio QA benchmark spanning multiple reasoning regimes.** We curate a synthetically generated large-scale dataset covering heterogeneous medical audio sources, including non-speech physiological sounds and conversational speech. The benchmark supports answering 46,701 questions requiring skills such as short-context inference, long-context temporal reasoning, multi-turn conversational reasoning, and reasoning about physiological sounds enabling systematic evaluation of audio-only and multimodal reasoning models.

- **A scalable synthetic audio generation pipeline.** We propose a controlled generation framework for synthetic medical speech that embeds physiological artifacts (e.g., coughs), emotional and distress markers, and temporally structured information. This design enables scalable dataset construction while preserving explicit control over reasoning complexity.

- **A reasoning-focused evaluation protocol for generative audio models.** Beyond multiple-choice QA, we include open-ended and voice-embedded QA settings in which questions and answers are integrated directly into the audio waveform. Open-ended questions require unconstrained reasoning over extended audio context and concise answer generation, providing a stringent test of generative audio reasoning capabilities, while Voice-based QA pairs require reasoning in the presence abrupt context change from interpreting the clinical conversation to understanding the question.

- **Scalable generation of challenging benchmarks.** A core contribution of this work is establishing that synthetic QA generation can produce difficult, clinically grounded evaluation data at scale. Our pipeline generates 46,701 question–answer pairs with minimal human oversight, yet the resulting benchmark remains highly challenging: state-of-the-art models achieve only 68.1% (Gemini-2.5-Pro (et. al., 2025)), 60.5% (Gemini-2.5-Flash (et. al., 2025)), and 42.8% (Qwen-2.5-Omni-7B (Xu et al., 2025a)) weighted average accuracy. This validates synthetic generation as a

scalable paradigm for constructing rigorous domain-specific benchmarks.

## 2. Related Work

### 2.1. Existing Audio QA Benchmarks

Audio question answering (AQA) has become a central paradigm for evaluating reasoning over acoustic signals, extending QA beyond text to temporally structured audio. Early benchmarks such as Clotho-AQA (Lipping et al., 2022) and Audiopedia assess audio-language understanding and compositional reasoning, while scalable spoken-QA pipelines convert text-based resources into speech. Long-form AQA benchmarks, including CORAAL-QA (Shankar et al., 2024), emphasize multi-turn interaction and long-range temporal reasoning. Beyond AQA, holistic audio reasoning benchmarks such as MMAU , MMAR, AudioBench, MuChoMusic (Weck et al., 2024), MMSU (Anonymous, 2026), Beyond Single Audio (Chen et al., 2024), and Dynamic-SUPERB Phase-2 (yu Huang et. al., 2024) expand evaluation to multi-audio inputs, instruction following, cultural diversity, and large task suites, consistently revealing that current audio–language models struggle with multi-step, multi-source, and long-context reasoning.

### 2.2. Medical Audio QA and Reasoning

Medical audio question answering remains a nascent area. CaReAQA (Wang et al., 2025b) introduces QA tasks over clinical audio such as heart and lung sounds, combining acoustic diagnostics with language reasoning. The benchmark demonstrates the feasibility of reasoning over diagnostic cues embedded in physiological signals but is limited in scale and mainly focuses on short, isolated audio segments. Text-based medical QA datasets (e.g., MeDiaQA (Suri et al., 2021)) capture complex clinical reasoning but entirely abstract away acoustic information, preventing evaluation of reasoning over speech, auscultation, or other medical audio signals. Overall, there is a significant gap in large-scale, multimodal medical QA benchmarks that evaluate long-context reasoning, multi-turn interactions, and generalizable reasoning over diverse acoustic evidence.

### 2.3. Large Audio–Language Models (LaLMs) and Reasoning

Large audio–language models (LaLMs) integrate pretrained speech or audio encoders with powerful language models, enabling inference over temporally structured acoustic inputs rather than isolated acoustic events. A range of recent architectures exhibit emergent audio reasoning capabilities. Audio Flamingo extends the Flamingo (Alayrac et al., 2022) multimodal framework to audio–language inputs, demonstrating strong in-context learning and few-shot generalization over diverse audio signals. Related models such as SpeechGPT (Zhang et al., 2023), Qwen-Audio, and SALMONN (Tang et al., 2023) pursue unified speech–audio–language modeling, enabling instruction following and compositional reasoning over both spoken and non-speech audio. LTU-AS (Gong et al., 2023) and AudioPaLM further highlight the effectiveness of scaling language-centric models for audio understanding, while LARM advances this line of work by explicitly framing audio understanding as a reasoning problem, emphasizing long-context modeling, cross-modal abstraction, and structured inference over acoustic evidence. Despite these advances, models and benchmarks remain largely general-domain, relying on short audio segments and leaving open the challenge of evaluating reasoning over medical audio, where clinically meaningful cues are often subtle and embedded within long-form recordings or complex conversations.

## 3. Methodology

### 3.1. Question-Answer Pair Generation Pipeline

To rigorously evaluate audio reasoning models, MedMosaic considers a diverse set of cognitive and perceptual skills. It is designed as a structured framework to probe both low-level perceptual processing and high-level reasoning capabilities. The QA pairs in the dataset are generated using Gemini-3-flash. Except for open-ended questions, all questions include 10 carefully crafted options, with only one of them being correct. We generate three questions at increasing levels of difficulty: Easy, Medium, and Hard. We ensure that the questions and options are generated such that they cannot be answered by referring to the LLM's medical knowledge database without listening to the audio.

Incorrect options or distractors are designed to be close and contrastive to the correct answer, ensuring they are plausible yet subtly incorrect. They may reflect common pitfalls, such as over reliance on keyword matching while being temporally inconsistent or describing similar acoustic characteristics but having close but dissimilar interpretations. To maximize difficulty, words and phrases are repeated aggressively across options, increasing lexical similarity and obfuscation, while overt conversational cues are deliberately avoided. This design ensures that model success depends on both perceptual acuity and nuanced reasoning, rather than simple pattern matching.

### 3.2. Question Answer Pair Generation Types

#### 3.2.1. SOUND ONLY QUESTION-ANSWER PAIR FORMATION

For sound-only question answering, we focus exclusively on non-speech medical audio, restricting the current scope to heart sounds, lung sounds, and cough sounds. This set-

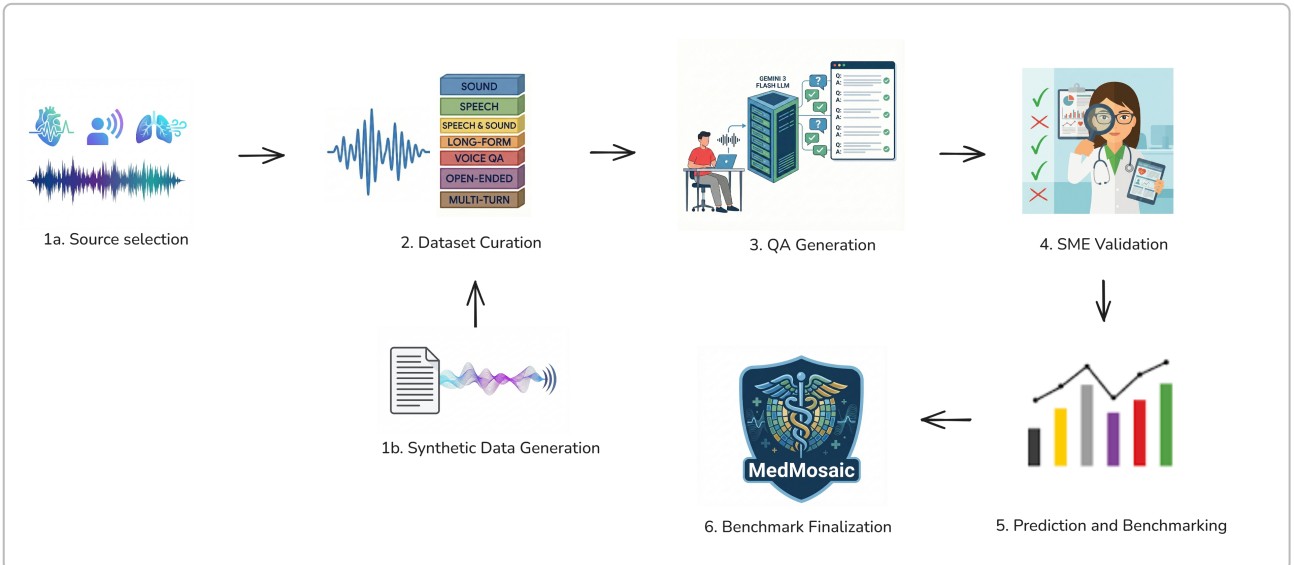

*Figure 2.* QA pair generation pipeline for MedMosaic.

ting removes linguistic cues entirely and requires models to reason solely from acoustic structure, temporal patterns, and signal characteristics. The questions in this category are designed to evaluate whether audio reasoning models can identify, differentiate, and reason over clinically relevant sound patterns without access to spoken context.

We perform a fine-grained categorization of medical sounds based on their acoustic properties and temporal structure. Lung sounds include wheezes, characterized by sustained, tonal, and narrowband energy, and crackles, which appear as brief, explosive, broadband transients. We also include stridor, a high-pitched, harsh, predominantly monophonic sound with strong tonal dominance, typically continuous during airflow through the upper airway. Cough sounds are subdivided into multiple clinically distinct acoustic classes, including wet coughs with bubbling or gurgling overlays, dry coughs with sharp, abrupt, non-resonant impulses, pertussis coughs marked by clustered repetitive bursts followed by a high-energy inspiratory intake ("whoop") and delayed recovery breathing, and barky coughs consisting of short, explosive events with low-pitched, hollow, resonant qualities. Heart-related sounds include heart murmurs, defined as sustained or semi-sustained tonal or noise-like energy occurring between primary heartbeats, with duration and intensity tied to the cardiac cycle rather than discrete impulses.

In constructing sound-only question–answer pairs, we place particular emphasis on the temporal structure and phasing of medical sounds rather than on isolated acoustic events. Questions are designed to require reasoning about when a sound occurs and how it aligns with underlying physiological cycles. The questions require models to reason about the coupling of acoustic events to physiological cycles, including the respiratory cycle (inhalation and exhalation) and the cardiac cycle (from S1 through systole to S2 and diastole), rather than relying on surface-level sound identification.

In addition to qualitative temporal reasoning, we include quantitative and count-based questions that require approximate estimation from audio alone. These questions involve counting or rate estimation over defined time windows, such as the number of breaths per unit time, cough frequency within a segment, heartbeat regularity or variability, and relative duration ratios between sound and silence.

### 3.2.2. SPEECH ONLY AUDIO QUESTION-ANSWER PAIR FORMATION

The question-answer pairs generated through the speech only prompt are designed to rigorously test a model's ability to comprehend clinical dialogues not solely through textual content, but through the auditory and conversational dialogues that carry diagnostic significance. To achieve this, we designed a prompt that generate adversarial MCQs from clinical dialogues under 3 mins in length. This layered difficulty structure systematically probes whether models genuinely comprehend clinical speech or merely exploit superficial lexical patterns. The core objective is to force models to attend to subtle conversational cues.

A central design constraint is the anti-hallucination principle: every correct answer must be derivable solely from what is audible in the recording, with no dependence on external medical knowledge. Additionally, each option must lead to a distinct clinical interpretation, preventing trivial elimination through surface-level reasoning. Collectively, our prompt design enables models to infer patient state,

*Table 1.* Accuracy of evaluated models on MedMosaic across different QA pair types (Sound Only, Speech + Sound, Short and Long Form Speech, Open Ended Speech, Open Ended Speech + Sound, Voice-Based and Multi-turn QA pairs) along with overall weighted averages. Bold values highlight the highest value in each row whereas underlined value highlights the highest value in a column.

| Model | Weighted Avg | MCQ_Long_Form | MCQ_Sound_Cough | MCQ_Sound_Heart | MCQ_Sound_Lung | MCQ_Speech | MCQ_Speech_Sound | Multi_Turn | OE_Speech | OE_Speech_Sound | Voice_QA |
|---|---|---|---|---|---|---|---|---|---|---|---|
| Audio-flamingo-3 | 24.1 | 8.1 | 26.2 | 37.8 | 37.9 | 10.7 | 12.6 | 26.7 | **55.2** | 37.9 | 0.1 |
| Audio-reasoner | 32.8 | 14.4 | 48.1 | 35.6 | 41.7 | 23.7 | 29.1 | 30.0 | **51.2** | 40.9 | 9.9 |
| Baichuan-omni | 38.6 | 36.8 | 43.6 | 26.6 | 33.0 | 43.5 | 32.8 | 53.9 | **57.6** | 47.5 | 31.5 |
| Desta25-audio | 41.0 | 24.9 | 20.2 | 37.1 | 44.6 | 49.4 | 41.2 | 39.4 | **56.0** | 45.4 | 9.1 |
| Gama | 23.2 | 11.4 | 33.6 | 36.6 | **39.1** | 12.7 | 12.9 | 35.0 | 38.1 | 29.1 | 8.9 |
| Gemini-2-5-flash | 60.5 | 73.6 | 52.8 | 47.2 | 51.6 | 66.6 | 60.2 | **78.3** | 74.8 | 68.2 | 52.7 |
| Gemini-2-5-pro | 68.1 | 67.4 | 67.0 | 61.7 | 52.8 | 68.0 | 68.3 | 76.7 | **79.2** | 73.5 | 70.3 |
| Gemma-3n-8b | 42.1 | 24.5 | 48.0 | 42.0 | 47.9 | 42.8 | 33.3 | 45.0 | **64.8** | 51.1 | 10.2 |
| Gpt4o-audio | 35.7 | 58.7 | 2.1 | 3.2 | 4.1 | 57.8 | 35.2 | **76.1** | 60.8 | 53.3 | 43.9 |
| Kimi-audio | 36.4 | 36.3 | 42.2 | 36.6 | 40.7 | 35.0 | 26.0 | **72.2** | 55.8 | 49.5 | 20.1 |
| Phi4-mm | 37.3 | 24.7 | 47.9 | 41.3 | 40.3 | 32.8 | 26.6 | **55.6** | 54.3 | 46.6 | 30.6 |
| Qwen-omni-7b | 42.8 | 39.0 | 48.4 | 33.8 | 36.6 | 52.7 | 32.5 | **65.0** | 57.0 | 47.6 | 29.9 |
| R1-AQA | 20.8 | 7.4 | **31.3** | 27.9 | **31.3** | 11.5 | 14.5 | 23.3 | 38.1 | 29.1 | 3.3 |

treatment implications, and risk trajectories.

### 3.2.3. SPEECH AND SOUND QUESTION-ANSWER PAIR FORMATION

Existing open-source medical audio datasets are largely limited to speech-only or isolated sound recordings, which fail to capture the full acoustic complexity of real clinical interactions. In natural medical conversations, involuntary and non-verbal sounds such as coughs, wheezes, sighs, and other physiological cues often carry critical diagnostic significance. The absence of these sounds results in datasets that lack important contextual information required for robust medical audio reasoning.

To address this limitation, we developed a synthetic audio generation pipeline that integrates realistic spoken dialogue with contextually appropriate acoustic events. Using the ElevenLabs v3(ElevenLabs, 2025) text-to-speech model, we generate high-fidelity medical audio that includes both verbal content and non-verbal acoustic cues. Raw transcripts from source datasets are first processed using the Qwen 2.5 14B(Xu et al., 2025a) Instruct model to enrich the text with acoustic placeholder tags, ensure natural conversational flow through pause indicators, and insert category-specific sounds such as respiratory, pain-related, or emotional cues. The resulting data are systematically organized into clinical categories and subcategories to reflect real-world medical diversity. These categories include, but are not limited to, Allergy and Dermatology, Cardiovascular, Respiratory, Gastrointestinal, Musculoskeletal, Neurological, and Genitourinary conditions. To further ensure naturalness and diversity in the generated audio, we manually curated a pool of 151 voices from the ElevenLabs voice library. Voices were selected based on different speaker roles, demographic diversity, and acoustic suitability for medical contexts.

### 3.2.4. MULTI TURN QUESTION-ANSWER PAIR FORMATION

We construct multi-turn(QA) sequences with explicit answer-conditioned dependencies across 3 turns. Incorrect answers to earlier questions introduce ambiguity, preventing shortcut reasoning. To maintain contextual continuity without leaking answers, each successive question refers to preceding information using anaphoric identifiers such as 'this', 'given this', 'from this', 'with this', 'here', or 'from now on'. Positional cues such as 'above', 'previous', or 'prior' are explicitly avoided. This setup encourages models to maintain and update an internal state across turns and perform multi-step reasoning rather than isolated question answering.

Beyond structural dependencies, the multi-turn QA sequences are designed to include progressive symptom reasoning, in which an initial report of discomfort or indirect symptom mention is gradually interpreted using causal cues, nonverbal evidence, or changes in frequency, leading to an assessment of severity, dominance, or risk trajectory. In addition, the QA pair sequence emphasizes multi-signal integration across turns by combining verbal symptoms, physiological cues, and behavioral indicators to update risk or detect escalation despite compliance. In the last turn, long-range reasoning is emphasized by probing latent information deduction. In this turn, information established implicitly through correct answers in earlier exchanges, and referenced only indirectly, is integrated with a hypothetical yet clinically plausible assumption to form questions whose answers are collectively implied by the prior conversational context, rather than being explicitly stated or directly observable in the input audio.

### 3.2.5. LONG FORM AUDIO QUESTION-ANSWER PAIR FORMATION

For long-form audio, we focus on extended clinical recordings of at least 3 minutes in duration, consisting of multiple conversational turns between a clinician and a patient, with diversity in patient gender and the medical issues discussed. These recordings are designed to reflect realistic clinical encounters in which clinically relevant information is distributed across time, rather than localized to a single contiguous segment. Consequently, questions in this category require models to retrieve and integrate evidence across distant dialogue turns, often separated by several minutes of

intervening conversation.

A central challenge in this setting is multi-hop temporal reasoning over audio, where correct inference depends on chaining together multiple pieces of evidence that are individually insufficient but collectively decisive. In particular, we emphasize delayed evidence usage, in which an early symptom report or instruction becomes relevant only after a later dialog introduces new context, clarification, or diagnostic information. To ensure that performance reflects genuine long-range reasoning rather than surface-level pattern matching or keyword spotting, the questions do not restate or explicitly reference the earlier evidence.

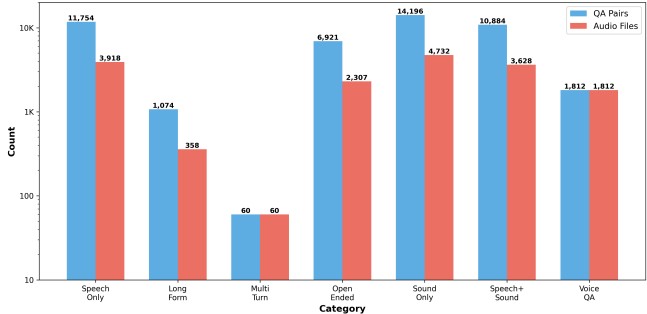

*Figure 3.* Distribution of QA pairs and audio files across MedMosaic categories.

### 3.2.6. VOICE BASED QUESTION ANSWER PAIR GENERATION

Existing audio understanding benchmarks predominantly evaluate models through text-based question–answer interfaces, where reasoning occurs over textual queries while audio serves merely as input. This paradigm fails to assess whether models can comprehend spoken instructions delivered within the audio modality itself a capability essential for real-world deployment in voice-activated clinical systems. Voice-based evaluation introduces unique challenges like segmenting continuous audio and integrating temporally distant information. We construct the Voice QA evaluation set from approximately 15% of the Speech+Sound category, encompassing naturalistic clinical dialogues, physiological sounds (coughs, wheezing, labored breathing), and paralinguistic phenomena (hesitations, vocal affect, disfluencies). The pipeline transforms text-based MCQs into vocalized segments appended to source audio using high-fidelity ElevenLabs TTS model with structured templates and silence intervals. Audio reasoning models must process a single continuous stream containing clinical conversation and vocalized questions, requiring acoustic segmentation, extended context retention, and integration of verbal and non-verbal evidence.

### 3.2.7. OPEN ENDED QUESTION-ANSWER PAIR FORMATION

Open ended question answering is an important paradigm for evaluating audio understanding and reasoning in large audio language models, as it requires generating explanations rather than choosing from fixed options. Despite this, open ended QA grounded in medical speech and in speech with accompanying sounds remains limited in existing benchmarks. To address this gap, we design an open ended QA framework grounded in medical speech and speech plus sound audio. Questions are framed to appear broadly clinical, emphasizing elaboration of severity, risk, or implications, while requiring attention to specific and subtle conversational cues to answer correctly. Each question combines

multiple aspects, such as evidence and interpretation or stated claims and implicit contradictions, so that the answer requires synthesis rather than recall. The questions do not explicitly name the cues they rely on instead, correct responses must infer meaning from hesitation, qualified/absolute statements, pauses, or emphasis. Interpretive formulations such as "What does X suggest or indicate about Y" are used to focus on clinical implications rather than literal content. Answers are open ended but tightly constrained in length and focus on accuracy rather than verbosity. By requiring description of delivery level evidence, the open ended QA pairs capture reasoning abilities that multiple choice formats may miss, since the possibilities are constrained by answer choices.

The prompts for generating QA pairs for each QA pair type can be found in Appendix A.11

### 3.3. Datasets

In this work, we leverage a diverse set of open-source datasets to comprehensively evaluate audio-based reasoning across modalities. Speech-only QA is drawn from Ekacare, MultiMed, Primock Short, and long-form Primock 57(Ekacare, 2024; Le-Duc et al., 2025; Korfiatis et al., 2022; Na0s, 2024), with the latter supporting multi-turn, long-form interactions essential for temporal reasoning. Sound-only QA uses CoughVID(Orlandic et al., 2021) , HLS-CMDS(Torabi et al., 2025) , and the Circor Digiscope Phonocardiogram datasets(Oliveira et al., 2022), capturing clinically relevant acoustic events. For speech-plus-sound and voice QA, we employ MTS Dialog and Kaggle datasets(har1, 2024), enabling cross-modal integration. To further stress test reasoning capabilities, we also generate a synthetic dataset (Appendix A.6). Collectively, these datasets provide a complementary and challenging benchmark suite, encompassing diverse modalities, conversation lengths, and clinical scenarios (Appendix A.1).

### 3.4. Subject Matter Expert (SME) Validation

To assess the clinical accuracy of synthetically generated QA pairs, we conducted validation with two healthcare professionals serving as subject matter experts. We employed stratified sampling to select 145 QA pairs ensuring representation across all categories, audio modalities, and difficulty levels. QA pairs were distributed across SMEs without overlap to maximize validation coverage. SMEs evaluated each QA pair across multiple dimensions using a 3-point scale and provided a final verdict. Results indicate strong clinical validity: 72.4% were accepted without modification, 22.8% required minor revisions (e.g., rewording the question, semantic phrasing in distractor options, response too detailed for audio content), and only 4.8% were rejected (e.g., duplicate distractor wording, multiple options could be considered partially correct, answer contains minor inaccuracies). Further details on the SME validation process are provided in Appendix A.10.

## 4. Experiments

### 4.1. Experimental Setup

We compare a diverse set of open-source and closed-source large language models (LLMs) for audio reasoning, spanning multiple architectural families. These include LALM-based models such as Salmonn-13B, GAMA-7.4B, Phi-4-MM-5.5B (Abouelenin et al., 2025), Audio Flamingo-8.4B, and Gemma-3n-E4B-IT-7.5B (Team et al., 2025a); LaRM-style or explicitly audio-centric reasoning models such as R1-AQA-8.2B (Li et al., 2025a) and Audio Reasoner-8.4B (Xie et al., 2025); and unified multimodal foundation models including Qwen-2.5-Omni-7B–10.7B, GPT-4o Audio (OpenAI, 2024), and Gemini-2.5-Flash. A brief description of each model and its architectural design is provided in Appendix A.9. For evaluation, we report percentage accuracy for all structured audio QA pairs, while open-ended questions are assessed using an LLM-as-a-judge paradigm.

To evaluate MCQ responses, we treat the answer generated by Gemini-3-Flash as the ground truth and compare it against the output produced by each benchmarked model. The evaluation follows a string-matching procedure that is robust to variations in response formatting. First, the ground-truth answer is normalized to a standard representation using regular expressions (e.g., both "The answer is (c)." and "c" are mapped to "(c)"). Next, we parse the model's response to extract the predicted answer. Since model outputs vary across the 13 evaluated systems, we employ a hierarchical parsing strategy that begins with strict JSON parsing models are prompted to respond in JSON format and falls back to string-based parsing when necessary. This includes extracting values associated with common keys (e.g., final answer, answer) as well as handling cases where the answer

appears only within the reasoning text. The parser robustly handles patterns such as "The answer is (c)," "Final Answer: (c)," and "Answer: c." Finally, the extracted model answer is compared against the normalized ground truth using a case-insensitive exact match over alphabetic choices (a–z). A correct match is assigned a score of 1.0, and an incorrect match is assigned 0.0.

We employed GPT-5.1 (Singh et al., 2025) as the judge model to evaluate open-ended question-answering (QA) responses using a four-criterion framework, with correctness, relevance, completeness, and clarity each scored on a continuous 0.0–1.0 scale. Correctness measures factual alignment with the reference answer, relevance assesses whether the response directly addresses the question without extraneous information, completeness evaluates coverage of all key points in the reference answer, and clarity measures coherence and organization. The evaluation prompt instructs GPT-5.1 to compare each model-generated response against the ground-truth answer using the question as context while enforcing strict scoring rules: any factual contradiction caps the correctness score at 0.5, and missing key information incurs proportional completeness penalties. The final score is computed as the unweighted average of the four criteria, and the evaluator outputs detailed JSON-formatted results with per-criterion scores and justifications to support fine-grained analysis across multiple qualitative dimensions.

### 4.2. Results

Table 1 summarizes the results of our benchmarking experiments. The weighted-average metric aggregates category-wise accuracies weighted by the number of questions per category, providing a consolidated performance measure. Under this metric, Gemini-2.5-Pro outperforms all other evaluated models. Gemini-2.5-Pro achieves the highest accuracy across all evaluation categories except Long-Form and Multi-turn MCQ, where it is outperformed by Gemini-2.5-Flash. We hypothesize that this behavior is related to the larger parameter count of Gemini-2.5-Pro, which may favor short- and medium-horizon reasoning over extended answer generation. We also observe that 5 out of the 13 models we benchmark, achieve their best performance on Multi Turn QA pairs. We attribute the relatively higher accuracy to the fact that reasoning models can achieve better performance on later turns by deriving context from the initial turns.

Audio Flamingo 3 (AF3) (Goel et al., 2025) has recently attracted significant attention due to its unified representation learning framework spanning speech, sound, and music. However, its training data are predominantly drawn from domains such as formal or read speech (e.g., SGPISpeech (O'Neill et al., 2021), LibriSpeech (Panayotov et al., 2015)), audiobooks, musical audio (e.g., FMA (Defferrard et al., 2016), Music4All (Santana et al., 2020), MUSDB18 (Rafii

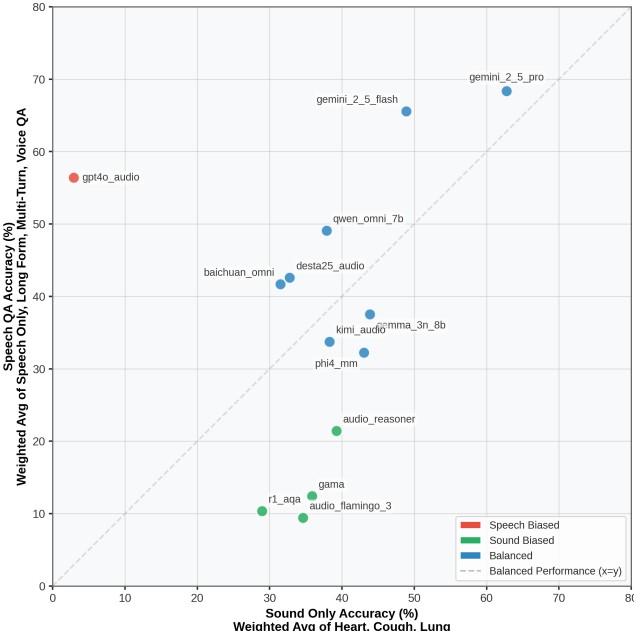

*Figure 4.* % Accuracy performance relative to speech only and sound only audio normalized by QA pair count . Omni models offer a balanced performance between the two catoegories, whereas GPT-4o-audio and Audio Flamingo 3 are biased towards speech only and sound only audio resp.

et al., 2017), MedleyDB-Pitch (Bittner et al., 2014)), and environmental sound datasets (e.g., Speech-in-SoundCaptions (Abu-El-Haija et al., 2016), MiraData (Ju et al., 2024), SoundDescs (Koepke et al., 2022)). During full fine-tuning and context-extension stages, the model further leverages AudioSkills-XL and LongAudio-XL. AudioSkills-XL augments AudioSkills with speech-in-sound audio from YouTube-8M (Abu-El-Haija et al., 2016), conversational audio from GigaSpeech (Chen et al., 2021), and read speech from LibriSpeech. Despite their diversity, these datasets do not adequately capture the acoustic and conversational characteristics of clinical environments. We argue that this domain mismatch contributes to the comparatively lower performance of AF3 on our benchmark. While LongAudio-XL models long form multi-speaker speech with datasets like Spotify Podcasts, Switchboard, Fisher (Cieri et al., 2004), MELD (Poria et al., 2019), DailyTalk (Lee et al., 2023) and MMDialog (Feng et al., 2023), none of them include doctor-patient meetings, which reflects in AF3's poor accuracy for Long Form MCQ's in Table 1.

### 4.3. Chain of Thought (CoT) Analysis

Motivated by the substantial performance gap between Gemini-2.5-Pro and GPT-4o-Audio on sound-based QA pairs, we analyze the reasoning traces produced by both models during inference. We observe that GPT-4o-Audio rejects a non-trivial fraction of audio inputs. In particular, it

rejects 1,729 out of 9,729 heart sound waveforms, 672 out of 3,884 cough sound waveforms, and 87 out of 585 lung sound waveforms, substantially more than any other model reported in Table 1. The inability of the GPT-4o-Audio model to comprehend a large proportion of physiological sounds primarily contributes to it's low accuracy percentages in the MCQ Sound categories. We analyze the CoT of Multi Turn QA pairs for the interdependence among turns and find a correlation as expected. For instance, in a sequence of 3 CoT's, the first detects spoken cues about "red, itchy skin" and incorrectly links it to workplace related stress instead of a swimming session. However, to argue about the correct answer in the second turn, the models' CoT makes an explicit reference to "dry, itchy skin", demonstrating that we are able to leverage partial information from the previous (incorrect) answer to infer the correct answer.

### 4.4. Limitations of our approach

A key limitation of our benchmarking methodology stems from the answer extraction procedure, which relies on parsing the model's chain-of-thought (CoT) and predicted response. Manual inspection of multi-turn CoT responses reveals that, in certain audio-unavailable scenarios, models may still produce answers by exploiting domain knowledge and lexical cues present in the question, rather than reasoning over the audio input. Due to the verbosity and variability of CoT outputs, our parsing strategy is not always able to reliably identify and filter such spurious answer generation. Consequently, when the question contains sufficient semantic cues to enable a correct guess, this behavior can artificially inflate measured performance, potentially affecting the accuracy reported in Table 1.

### 5. Conclusions

We introduced MedMosaic, a large-scale medical audio question-answering benchmark comprising 46,701 QA pairs across diverse clinical audio modalities, including physiological sounds, clinical conversations, and their combinations. The benchmark supports multiple reasoning paradigms such as short-context inference, long-form temporal reasoning, multi-turn conversational chains, and open-ended generation, enabling systematic evaluation of audio reasoning capabilities in medical settings.

Our evaluation of 13 models reveals three key findings. First, medical audio reasoning remains challenging: even Gemini-2.5-Pro achieves only 68.1% accuracy. Second, some models exhibit pronounced biases toward either speech or sound understanding, with few demonstrating balanced cross-modal competence. Third, substantial performance variation across question types highlights divergent model strengths in handling different reasoning demands. Our synthetic generation pipeline, validated by clinical experts with

72.4% acceptance, establishes a scalable approach for constructing domain-specific benchmarks. Limitations include potential gaps between synthetic and real clinical audio characteristics, and the current focus on English-language interactions.

## Impact Statement

This paper presents work whose goal is to advance the evaluation of audio reasoning models in medical settings. Our benchmark relies entirely on publicly available datasets and synthetic audio generation, avoiding direct patient data collection. While improved medical audio understanding could benefit clinical decision-making, we emphasize that benchmark performance does not indicate readiness for clinical deployment, and any real-world application would require extensive additional validation. We do not foresee immediate negative societal consequences beyond those well established when advancing machine learning for healthcare applications.

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

# A. Appendix

## A.1. Dataset Description

- **Primock57**(Korfiatis et al., 2022): It consists of 57 long mock medical primary care consultations held over 5 days by 7 Babylon clinicians and 57 Babylon employees acting as patients, using case cards with presenting complaints, symptoms, medical and general history etc.

- **Primock-med**(Korfiatis et al., 2022; Na0s, 2024): It consists of 322 short mock medical primary care consultations between 7 Babylon clinicians and 57 Babylon employees acting as patients. It is created by chunking longer consultation recordings from the original Primock57 audio corpus. Each audio segment is exactly 30 seconds in length.

- **Ekacare**(Ekacare, 2024): It includes approximately 3,600 English recordings and 320 Hindi recordings featuring medical terminology, including branded drugs specific to the Indian context, delivered across various speaking styles such as isolated medical entities, narrated medical sentences, and impromptu conversations.

- **MTS Dialog**(har1, 2024): It consists of 1,701 doctor-patient conversations paired with structured clinical note summaries. The dataset is divided into a training set of 1,201 conversation-summary pairs and a validation set of 100 pairs. Each clinical note follows a standardized format comprising four sections: Symptoms, Diagnosis, History of Patient, and Plan of Action, with "N/A" entries indicating missing information for specific sections.

- **CoughVID**(Orlandic et al., 2021): It is a large-scale crowdsourced cough audio dataset designed for respiratory condition analysis and COVID-19 screening. It contains over 25,000 cough recordings spanning diverse ages, genders, geographic regions, and COVID-19 statuses. The dataset includes an open-source cough detection algorithm to support data quality and robustness assessment. In addition, more than 2,800 recordings were expert-annotated by four physicians for medically relevant cough abnormalities, making it one of the largest clinically labeled cough datasets available. To enhance label reliability, symptomatic and COVID-19 coughs were collected from regions with high infection rates.

- **HLS-CMDS (Heart and Lung Sounds Dataset Recorded from a Clinical Manikin using Digital Stethoscope)**(Torabi et al., 2025): It consists of 535 heart and lung sound recordings collected from a clinical manikin using a digital stethoscope. It includes isolated heart sounds (50), isolated lung sounds (50), and mixed cardiopulmonary recordings (145), with each mixed recording paired with its corresponding source heart and lung signals (145 each). Recordings span multiple auscultation locations and include both normal and pathological conditions. Representative sound types include normal heart sounds, systolic and diastolic murmurs, atrial fibrillation, wheezing, and crackles, captured at clinically relevant landmarks such as the upper sternal borders, apex, and anterior lung fields.

- **MedDialog-Audio**(Gassenn et al., 2025): It consists of 147,476 audio files derived from the MedDialog-EN transcription corpus, specifically designed for automatic speech recognition research in healthcare settings. Unlike naturally recorded audio, this dataset employs a synthetic generation pipeline comprising three stages: text normalization of original transcriptions using language models, speech synthesis via text-to-speech conversion.

- **CirCor DigiScope Phonocardiogram Dataset 1.0.3**(Oliveira et al., 2022): It comprises 5,272 heart sound recordings collected from the four primary auscultation sites of 1,568 subjects aged 0–21 years (mean ± SD: 6.1 ± 4.3). Recordings range from 4.8 to 80.4 s in duration (mean ± SD: 22.9 ± 7.4), totaling over 33.5 hours. Each cardiac murmur is exhaustively annotated by experts for timing, shape, pitch, grade, quality, and anatomical location. In addition, S1 and S2 heart sound boundaries are provided via a semi-supervised segmentation procedure.

- **Hani89**(Hani89, 2024):It contains 6,661 audio utterances for common medical symptoms like "knee pain" or "headache,". Each utterance was created by individual human contributors based on a given symptom.

- **MultiMed**(Le-Duc et al., 2025): It contains 48,369 audio files with human-annotated transcripts across five languages (Vietnamese, English, French, German, and Chinese), sourced from real-world medical conversations published by professional medical channels on YouTube. Unlike simulated datasets where participants role-play as doctors and patients, MultiMed captures authentic medical discourse across diverse contexts. It encompasses 10 distinct recording conditions (Documentary, Interview, Lecture, News, Podcast, Webinar, Speech, Talk, Vlog, and Workshop) and features 6 speaker roles (Lecturer, Doctor, Host, Patient, Podcaster, and Broadcaster). The collection exhibits substantial linguistic diversity with representations from 16 different accents, providing a comprehensive resource for evaluating audio understanding in varied real-world medical communication scenarios.

*Table 2.* Summary of constituent datasets generating QA pairs for MedMosaic

| Category | Dataset Name | QA Pairs generated | Audio files processed |
|---|---|---|---|
| Speech Only | Primock-med | 855 | 285 |
| Speech Only | ekacare | 780 | 260 |
| Speech Only | Hani89 | 396 | 132 |
| Speech Only | MultiMed | 3792 | 1264 |
| Speech Only | MedDialog-Audio | 5931 | 1977 |
| Long Form | Primock57 | 171 | 57 |
| Long Form | MultiMed | 9 | 3 |
| Long Form | Synthetic MTS Dialog | 894 | 298 |
| Multi Turn | Primock57 | 57 | 57 |
| Multi Turn | MultiMed | 3 | 3 |
| Open Ended (speech) | Primock-med | 99 | 33 |
| Open Ended (Speech) | ekacare | 108 | 36 |
| Open Ended (Speech) | MultiMed | 5499 | 1833 |
| Open Ended (Speech+Sound) | Synthetic MTS Dialog | 1215 | 405 |
| Sound Only | CoughVID, HLS-CMDS | 3882 | 1294 |
| Sound Only | HLS-CMDS | 585 | 195 |
| Sound Only | HLS-CMDS, CirCor | 9729 | 3243 |
| Speech+Sound | Synthetic MTS Dialog | 10884 | 3628 |
| Voice QA | Synthetic MTS Dialog | 1812 | 1812 |
| Total | - | 46701 | 16815 |

## A.2. Example Questions

Table 3 presents representative questions from each QA category in MedMosaic, demonstrating the diversity and clinical complexity of our benchmark.

*Table 3.* Example QA pairs from MedMosaic across different question types.

| QA Type | Question | Options | Ans |
|---|---|---|---|
| Sound (Heart) | Based on the rhythmic identification of the S1 and S2 cardiac anchors, where is the abnormal acoustic energy located in the cardiac cycle? | (a) In the systolic interval between S1 and S2; (b) In the diastolic interval between S2 and the next S1; (c) Preceding the S1 heart sound in late diastole; (d) Overlapping S1 and continuing into early diastole; (e) Throughout both the systolic and diastolic intervals; (f) Only at the onset of the S2 heart sound; (g) Within the silent period following the S2 heart sound; (h) Intermittently between S1 and the next S1; (i) Isolated strictly to the mid-diastolic phase; (j) No abnormal energy is detected in the cycle | (a) |
| Sound (Cough) | What is the primary acoustic character of the coughing events observed in this recording? | (a) Single, isolated cough with a low-pitched, barking resonance; (b) Frequent, loose coughs with a distinct low-frequency bubbling sound; (c) Repetitive, clustered bursts with a dry, sharp quality followed by a sharp inhalation; (d) Abrupt, non-resonant dry coughs occurring at regular intervals; (e) Weak, suppressed coughing efforts with prolonged expiratory wheeze; (f) Rapid sequence of wet coughs followed by a quiet, effortless breath; (g) Deep, hollow barking coughs without an inspiratory sound; (h) Single explosive cough with a prolonged recovery phase; (i) Intermittent dry coughs with no clustered paroxysmal structure; (j) Sharp, non-resonant coughs followed by low-frequency gurgling | (c) |
| Sound (Lung) | Based on the provided audio, how are the primary adventitious sounds best categorized? | (a) Continuous, musical, high-pitched sounds; (b) Discontinuous, explosive, high-pitched sounds; (c) Continuous, non-musical, low-pitched sounds; (d) Discontinuous, non-musical, low-pitched sounds; (e) Prolonged absence of any respiratory sounds; (f) High-pitched, harsh sounds heard on inspiration only; (g) Low-pitched, grating sounds heard during inspiration; (h) Rhythmic, clicking sounds heard during expiration; (i) Sudden, gasping sounds followed by silence; (j) Normal, soft, rustling sounds of healthy lungs | (a) |
| Speech | What mechanism is described as the primary source of collarbone discomfort during respiration? | (a) Minimal accessory muscle use during coughing reflects stable clavicle health; (b) Minimal primary muscle use during coughing reflects stable clavicle health; (c) Limited accessory muscle use during coughing suggests localized clavicle inflammation; (d) Limited primary muscle use during coughing suggests localized clavicle inflammation; (e) Excessive primary muscle use during coughing signifies mechanical clavicle strain; (f) Moderate primary muscle use during coughing signifies mechanical clavicle strain; (g) Excessive accessory muscle use during coughing signifies mechanical clavicle strain; (h) Moderate accessory muscle use during coughing signifies mechanical clavicle strain; (i) Intense primary muscle use during coughing signifies mechanical clavicle strain; (j) Intense accessory muscle use during coughing signifies mechanical clavicle strain | (g) |
| Multi-Turn | What specific detail about the patient's abdominal pain suggests a functional or infectious motility issue rather than a localized, persistent inflammatory process? | (a) Pain is in the upper right quadrant; (b) Pain is constant and progressively worsening; (c) Pain radiates toward the patient's back; (d) Pain is a steady, burning sensation localized behind the navel; (e) Pain is a muscle-like cramp that fluctuates in intensity; (f) Pain is significantly relieved immediately after eating; (g) Pain is exclusively present during the act of defecation; (h) Pain is sharp, stabbing, and prevents deep breathing; (i) Pain is located primarily in the chest and throat area; (j) Pain is accompanied by a visible skin rash on the abdomen | (e) |
| Speech+Sound | What do the patient's vocal pauses reveal about their medical history report? | (a) Firm vocal delivery suggests reliable recall of specific medical history details; (b) Rapid vocal pace indicates certain recall of specific medical history details; (c) Steady vocal rhythm suggests accurate recall of specific medical history details; (d) Resonant vocal quality reveals confident recall of specific medical history details; (e) Fluent vocal articulation indicates robust recall of specific medical history details; (f) Assertive vocal tone suggests definitive recall of specific medical history details; (g) Clear vocal clarity reveals precise recall of specific medical history details; (h) Direct vocal response indicates immediate recall of specific medical history details; (i) Hesitant vocal pauses indicate uncertain recall of specific medical history details; (j) Prompt vocal timing suggests clear recall of specific medical history details | (i) |

## A.3. Success  Failure Cases

Table 4 illustrates representative examples where models correctly and incorrectly answered questions, highlighting the reasoning challenges posed by MedMosaic across different question types.

*Table 4.* Example model predictions across different question types. ✓ indicates correct prediction, ✗ indicates incorrect prediction.

| Question Type | Question | GT | Options | Model | Prediction |
|---|---|---|---|---|---|
| Sound | Based on the rhythmic identification of the S1 and S2 cardiac anchors, where is the abnormal acoustic energy located in the cardiac cycle? | (a) | (a) In the systolic interval between S1 and S2; (b) In the diastolic interval between S2 and S1; (c) Preceding the S1 heart sound; ... (j) No abnormal energy detected | Gemini 2.5 Pro | (j) ✗ |
| Speech | What mechanism is described as the primary source of collarbone discomfort during respiration? | (g) | (a) Minimal accessory muscle use during coughing; (b) Minimal primary muscle use during coughing; ... (g) Repetitive accessory muscle strain | GPT-4o | (g) ✓ |
| Multi-Turn | Based on the patient's report, which combination of physical findings and recent activities most likely contributed to their current presentation? | (c) | (a) Localized tenderness with recent exercise; (b) Diffuse pain with dietary changes; (c) Postural symptoms with prolonged sitting; (d) Radiating discomfort with lifting activity | GPT-4o | (a) ✗ |
| Long-Form | What does the conversational evidence specifically reveal about the patient's chest pain? | (d) | (a) Sharp pain radiating to the arm; (b) Dull pressure radiating to the arm; (c) Dull pressure with jaw discomfort; ... (j) Sharp pain radiating to the back | Gemini 2.5 Pro | (c) ✗ |

## A.4. No Audio Baseline Performance

A natural concern with any audio QA benchmark is whether the audio is actually needed, or whether enough signal leaks into the question and answer options that a language-only model can solve items without ever hearing the waveform. To check this for MedMosaic, we run the benchmark text-only: each model sees the question (and MCQ options where applicable) with no audio attached. We evaluate three models chosen to span architectures and performance tiers: Gemini-2.5-Flash (our second-strongest model overall at 60.5% weighted accuracy, and same family as Gemini-2.5-Pro, which we omit on cost grounds), Qwen-2.5-Omni-7B (the strongest open-source model at 42.8%, with a distinct architectural lineage), and Audio Flamingo 3 (sound-biased and lowest-tier at 24.1%, included to check whether weaker audio models were already near a text-only guessing regime).

Table 5 reports per-category accuracy under the no-audio condition. The weighted-average accuracies are 38.1% for Gemini-2.5-Flash, 30.8% for Qwen-2.5-Omni-7B, and 21.0% for Audio Flamingo 3. In every case, removing the audio causes a substantial drop relative to the audio-enabled numbers reported in the main paper, confirming that audio access materially contributes to model performance and that MedMosaic is not trivially solvable from text alone.

*Table 5.* Performance comparison across evaluation categories.

| Model | WeightedAvg | MCQ_Long_Form | MCQ_Sound_Cough | MCQ_Sound_Heart | MCQ_Sound_Lung | MCQ_Speech | MCQ_Speech_Sound | Multi_Turn | OE_Speech | OE_Speech_Sound | Voice_QA |
|---|---|---|---|---|---|---|---|---|---|---|---|
| Audio-flamingo-3 | 21.0 | 8.8 | 26.3 | 28.1 | 25.6 | 9.2 | 15.1 | 6.7 | 43.5 | 35.5 | 8.8 |
| Qwen-omni-7b | 30.8 | 11.4 | 46.1 | 42.5 | 63.3 | 11.5 | 32.9 | 40.8 | 29.6 | 25.5 | 0.0 |
| Gemini-2-5-flash | 38.1 | 36.1 | 32.3 | 51.2 | 43.9 | 30.4 | 28.9 | 47.8 | 55.7 | 37.8 | 9.9 |

## A.5. Difficulty-Stratified Performance

MedMosaic assigns each QA pair one of three difficulty levels, easy, medium, and hard, reflecting the depth of reasoning required to answer correctly given the accompanying audio. To characterize how model performance varies with item difficulty, we report per-category accuracy separately for each difficulty stratum across all 13 evaluated models. Tables 6a, 6b, and 6c present results for easy, medium, and hard items respectively. Across nearly all models and categories, accuracy decreases monotonically as difficulty increases, confirming that the assigned difficulty labels correspond to genuine increases in reasoning load. The gap between strong and weak models is preserved across difficulty levels, indicating that difficulty stratification does not collapse model rankings at either extreme.

*Table 6.* Model accuracy (%) stratified by evaluation category across three difficulty levels. Values are percentages over parseable responses; refused or malformed outputs are excluded.

*(a)* **Easy** QA pairs.

| Model | MCQ_Long_Form | MCQ_Sound_Cough | MCQ_Sound_Heart | MCQ_Sound_Lung | MCQ_Speech | MCQ_Speech_Sound | OE_Speech | OE_Speech_Sound | Voice_QA |
|---|---|---|---|---|---|---|---|---|---|
| Audio-flamingo-3 | 10.0 | 28.7 | 40.0 | 40.1 | 12.2 | 15.2 | 61.5 | 44.3 | 0.1 |
| Audio-reasoner | 16.1 | 51.9 | 31.8 | 44.1 | 26.0 | 30.9 | 56.0 | 45.3 | 11.3 |
| Baichuan-omni | 38.3 | 49.7 | 19.5 | 28.2 | 46.1 | 34.4 | 63.6 | 52.2 | 33.6 |
| Desta2.5-audio | 26.8 | 22.7 | 38.5 | 47.1 | 51.0 | 43.7 | 62.3 | 51.4 | 11.4 |
| Gama | 13.5 | 35.5 | 38.4 | 41.6 | 15.2 | 15.1 | 42.2 | 31.1 | 11.5 |
| Gemini-2.5-flash | 75.4 | 55.1 | 49.7 | 53.2 | 68.9 | 61.8 | 81.7 | 73.5 | 55.3 |
| Gemini-2.5-pro | 69.5 | 69.0 | 63.4 | 42.6 | 70.0 | 70.2 | 85.3 | 78.1 | 72.2 |
| Gemma-3-8B | 25.9 | 50.2 | 43.8 | 49.9 | 42.2 | 31.2 | 70.8 | 57.8 | 11.7 |
| Gpt4o-audio | 60.9 | 1.7 | 1.0 | 0.0 | 59.8 | 37.2 | 69.3 | 60.5 | 45.3 |
| Kimi-audio | 38.7 | 44.8 | 38.6 | 42.7 | 37.0 | 28.0 | 62.5 | 56.7 | 22.1 |
| Phi4-mm | 26.4 | 50.3 | 43.7 | 32.8 | 34.3 | 28.1 | 60.3 | 51.8 | 32.8 |
| Qwen-omni-7b | 41.0 | 50.0 | 35.3 | 38.9 | 54.6 | 34.3 | 63.7 | 52.9 | 31.4 |
| R1-AQA | 9.9 | 33.4 | 29.7 | 33.6 | 14.1 | 16.3 | 41.3 | 31.6 | 4.6 |

*(b)* **Medium** QA pairs.

| Model | MCQ_Long_Form | MCQ_Sound_Cough | MCQ_Sound_Heart | MCQ_Sound_Lung | MCQ_Speech | MCQ_Speech_Sound | OE_Speech | OE_Speech_Sound | Voice_QA |
|---|---|---|---|---|---|---|---|---|---|
| Audio-flamingo-3 | 8.1 | 26.2 | 37.8 | 37.9 | 10.7 | 12.6 | 55.4 | 38.6 | 0.1 |
| Audio-reasoner | 14.4 | 44.6 | 38.1 | 41.7 | 23.7 | 29.1 | 51.2 | 38.7 | 9.9 |
| Baichuan-omni | 36.8 | 39.6 | 28.2 | 33.9 | 43.5 | 32.8 | 57.0 | 45.9 | 31.5 |
| Desta2.5-audio | 24.9 | 20.2 | 37.1 | 44.6 | 49.4 | 41.2 | 55.1 | 42.8 | 9.1 |
| Gama | 11.4 | 33.6 | 36.6 | 39.1 | 12.7 | 12.9 | 37.2 | 28.8 | 8.9 |
| Gemini-2.5-flash | 73.6 | 52.8 | 47.2 | 51.6 | 66.6 | 60.2 | 74.2 | 66.2 | 52.7 |
| Gemini-2.5-pro | 67.4 | 67.0 | 61.7 | 57.9 | 68.0 | 68.3 | 78.5 | 71.0 | 70.3 |
| Gemma-3n-8b | 24.5 | 48.0 | 42.0 | 47.9 | 42.5 | 32.2 | 64.3 | 47.6 | 10.2 |
| Gpt4o-audio | 58.7 | 1.9 | 2.3 | 2.0 | 57.8 | 35.2 | 59.5 | 50.5 | 43.9 |
| Kimi-audio | 36.3 | 42.2 | 36.6 | 40.7 | 35.0 | 26.0 | 55.7 | 47.6 | 20.1 |
| Phi4-mm | 24.7 | 47.9 | 41.3 | 44.1 | 32.8 | 26.6 | 53.9 | 45.3 | 30.6 |
| Qwen-Omni-7b | 39.0 | 48.4 | 33.8 | 36.6 | 52.7 | 32.5 | 56.4 | 45.8 | 29.9 |
| R1-AQA | 7.4 | 31.3 | 27.9 | 31.3 | 11.5 | 14.5 | 38.5 | 29.1 | 3.3 |

*(c)* **Hard** QA pairs.

| Model | MCQ_Long_Form | MCQ_Sound_Cough | MCQ_Sound_Heart | MCQ_Sound_Lung | MCQ_Speech | MCQ_Speech_Sound | OE_Speech | OE_Speech_Sound | Voice_QA |
|---|---|---|---|---|---|---|---|---|---|
| Audio-flamingo-3 | 6.2 | 23.7 | 35.6 | 35.7 | 9.2 | 10.0 | 48.7 | 30.8 | 0.1 |
| Audio-reasoner | 12.7 | 48.5 | 37.7 | 39.3 | 21.4 | 27.3 | 46.4 | 38.8 | 8.5 |
| Baichuan-omni | 35.3 | 44.4 | 33.5 | 37.1 | 40.9 | 31.2 | 52.1 | 44.5 | 29.4 |
| Desta2.5-audio | 23.0 | 17.7 | 35.7 | 42.1 | 47.8 | 38.7 | 50.6 | 42.0 | 6.8 |
| Gama | 9.3 | 31.7 | 34.8 | 36.6 | 10.2 | 10.7 | 34.8 | 27.4 | 6.3 |
| Gemini-2.5-flash | 71.8 | 50.5 | 44.7 | 50.0 | 64.3 | 58.6 | 68.7 | 64.9 | 50.1 |
| Gemini-2.5-pro | 65.4 | 65.0 | 60.0 | 55.4 | 66.0 | 66.4 | 73.9 | 71.3 | 68.4 |
| Gemma-3n-8b | 23.1 | 45.8 | 40.2 | 45.8 | 43.1 | 33.6 | 59.4 | 47.9 | 8.7 |
| GPT4o-audio | 56.5 | 3.4 | 7.1 | 9.8 | 55.7 | 33.2 | 53.6 | 48.9 | 42.5 |
| Kimi-audio | 33.9 | 39.6 | 34.6 | 38.7 | 33.0 | 24.0 | 49.1 | 44.3 | 18.1 |
| Phi4-mm | 23.0 | 45.5 | 38.9 | 44.1 | 31.3 | 25.1 | 48.8 | 42.8 | 28.4 |
| Qwen-Omni-7b | 37.0 | 46.7 | 32.3 | 34.3 | 50.8 | 30.6 | 51.0 | 44.1 | 28.4 |
| R1-AQA | 4.9 | 29.2 | 26.1 | 29.0 | 8.9 | 12.7 | 34.6 | 26.6 | 2.0 |

## A.6. Synthetic Audio Generation

In this appendix, we present some more quantitative metrics relating to the generated synthetic audio. Tables 7, 8, 9 and 10 give the breakup of generated audio according to the role played in the conversation. Each table also features the age and gender based breakup within the role served.

The initial corpus comprised 4,573 medical conversations. After applying systematic curation criteria, the final dataset consists of 4,331 conversations, corresponding to the removal of 242 instances (5.2% reduction). Conversations were excluded if they contained insufficient medical information, were predominantly generic or conversational in nature, or lacked adequate clinical relevance to support the generation of at least easy-level medical comprehension question–answer pairs. This filtering process was designed to ensure that every retained conversation contains sufficient medically grounded content to enable meaningful and evaluative QA generation for benchmarking medical reasoning models.

*Table 7.* Doctor Voices

| Demographic | Count |
|---|---|
| Male, Middle Aged | 14 |
| Male, Young | 7 |
| Male, Old | 3 |
| Female, Middle Aged | 3 |
| Female, Young | 3 |
| Female, Old | 1 |

*Table 8.* Patient Voices

| Demographic | Count |
|---|---|
| Male, Middle Aged | 42 |
| Male, Young | 16 |
| Male, Old | 14 |
| Female, Middle Aged | 17 |
| Female, Young | 9 |
| Female, Old | 2 |

*Table 9.* Guest Family Voices

| Demographic | Count |
|---|---|
| Male, Young | 4 |
| Male, Middle Aged | 3 |
| Female, Young | 3 |

*Table 10.* Guest Clinician Voices

| Demographic | Count |
|---|---|
| Male, Middle Aged | 5 |
| Female, Young | 2 |
| Male, Young | 2 |
| Male, Old | 1 |

### A.7. Acoustic Tag List and Tag Placement Pipeline

To generate synthetic medical audio that faithfully reflects the acoustic complexity of real clinical encounters, we define a set of 35 acoustic tags that encode clinically relevant and paralinguistic events. Of these, 22 tags have direct medical relevance (e.g., respiratory sounds, pain indicators, physiological reflexes), while the remaining 13 capture conversational and emotional cues (e.g., hesitations, sighs, changes in speaking pace) that carry diagnostic significance in clinical communication.

During synthetic audio generation, raw transcripts from source datasets are processed by the Qwen 2.5 14B Instruct model, which is prompted to enrich each transcript by inserting acoustic tags at contextually appropriate locations. The model is instructed to preserve natural conversational flow, ensure that tag placement aligns with the clinical narrative, and avoid over-insertion that would compromise audio realism. Pause indicators are also inserted to model natural turn-taking and hesitation patterns observed in real doctor–patient interactions.

The enriched transcripts are then passed to the ElevenLabs v3 text-to-speech model, which interprets the embedded tags and renders the corresponding acoustic events within the generated audio. This two-stage approach using LLM-driven tag placement followed by TTS rendering enables scalable generation of medically grounded audio while maintaining explicit control over the type, frequency, and placement of acoustic cues.

The complete set of 35 tags is listed in Table 11. The tag placement prompt used to instruct Qwen 2.5 14B is also provided.

*Table 11.* Complete Taxonomy of 35 Unique Acoustic Tags

| Tag | Medically Relevant | Clinical Relevance | Description / Clinical Signal |
|---|---|---|---|
| pain_groan | Yes | Psychophysiological | Vocalization during active pain; acoustic marker of nociceptive distress |
| wince | Yes | Psychophysiological | Sharp, sudden pain reaction; marker of acute pain episodes |
| grimace | Yes | Psychophysiological | Sustained discomfort; marker of chronic or intense pain |
| cough | Yes | Diagnostic | Respiratory symptom; informative for URI, bronchitis, pneumonia, TB, COVID, asthma |
| sneeze | Yes | Diagnostic | Allergic or upper-respiratory marker; rhinitis, common cold, hay fever |
| wheezing | Yes | Diagnostic | Lower-airway obstruction; asthma, COPD, bronchospasm |

| Tag | Medically Relevant | Clinical Relevance | Description / Clinical Signal |
| --- | --- | --- | --- |
| clearing_throat | Yes | Diagnostic | Throat irritation, post-nasal drip, laryngopharyngeal reflux |
| heavy_breath | Yes | Diagnostic | Labored breathing, dyspnea, exertional intolerance |
| choking | Yes | Diagnostic | Aspiration, airway obstruction, dysphagia — acute clinical event |
| gasp | Yes | Diagnostic / Psychophysiological | Sudden inhalation from pain, shock, or air hunger |
| stutter | Yes | Psychophysiological / Neurological | Speech disfluency from stress, anxiety, or neurological condition |
| trembling_voice | Yes | Psychophysiological / Neurological | Vocal tremor from fear, distress, or neurological tremor disorders |
| rapid_breathing | Yes | Diagnostic | Tachypnea; anxiety, fever, sepsis, metabolic acidosis, cardiopulmonary distress |
| hoarse_voice | Yes | Diagnostic | Laryngitis, vocal strain, prolonged illness, sore throat, reflux |
| sniffle | Yes | Diagnostic | Nasal congestion, runny nose, post-nasal drip; allergies, URI |
| shiver | Yes | Diagnostic | Thermoregulatory response; fever chills, hypothermia, shock |
| nervous_breath | Yes | Psychophysiological | Anxiety, anticipatory worry, hyperventilation tendency |
| shaky_voice | Yes | Psychophysiological | Fear, severe worry, autonomic arousal |
| weak_voice | Yes | Psychophysiological | Fatigue, systemic illness, reduced respiratory drive |
| strained_voice | Yes | Psychophysiological | Speaking through pain, vocal effort under distress |
| heavy_sigh | Yes | Psychophysiological | Exhaustion, resignation, depressive affect, pain fatigue |
| relieved_sigh | Yes | Psychophysiological | Post-reassurance affective release; tension reduction |
| confused_pause | No | Communication | Comprehension hesitation; cognitive processing delay |
| hesitant_tone | No | Communication | Uncertainty in response; recall or commitment hesitation |
| grateful_tone | No | Communication | Affective closing cue; expression of thanks |
| pause | No | Communication | Generic prosodic filler; brief thinking moment |
| brief_pause | No | Communication | Very short hesitation; micro-prosodic filler |
| breath | No | Communication | Natural respiratory filler (non-pathological) |
| soft_tone | No | Communication | Neutral, calm vocal register |
| professional_tone | No | Communication | Formal clinical register; assessment-phase speech style |
| warm_voice | No | Communication | Rapport-building register; bedside-manner cue |
| concerned_tone | No | Communication | Affective response to patient distress |
| empathetic_voice | No | Communication | Acknowledgment of patient pain or worry |
| reassuring_voice | No | Communication | Comfort-delivery register; anxiety reduction |
| calm_voice | No | Communication | Procedure or explanation register; de-escalation cue |

```
acoustic_placeholder_prompt: |

You will receive a complete doctor-patient conversation transcript. Your task is to:
```

1. Understand the full medical and emotional context

2. Add contextual acoustic placeholders to each dialogue line

3. Extract explicit gender and age information

4. Output structured JSON with enriched transcript

---

## STEP 1: UNDERSTAND THE COMPLETE CONVERSATION CONTEXT

**Before processing anything, read the ENTIRE conversation to understand:**

- The medical condition and symptoms discussed

- The emotional progression throughout the conversation

- Pain levels and their evolution

- The doctor's approach (empathetic, professional, concerned)

- The patient's emotional state (anxious, in pain, confused, grateful)

- Any changes in tone as the conversation progresses

**CONVERSATION CONTEXT:**

{context_paragraph}

---

## STEP 2: ADD ACOUSTIC PLACEHOLDERS (INCREMENTALLY & CONTEXTUALLY)

### CRITICAL RULES (IN PRIORITY ORDER):

1. **MANDATORY: AT LEAST ONE PLACEHOLDER PER SENTENCE**: EVERY line of dialogue MUST have at least one placeholder. No exceptions. If there's no emotional context, use neutral placeholders like [pause], [breath], [soft_tone], [brief_pause]

2. **INCREMENTAL PROCESSING**: Add placeholders based on the CURRENT line in context of what has been said so far

3. **CONTEXTUAL AWARENESS**: Tags must reflect the ACTUAL content and medical situation being discussed

4. **NATURAL PLACEMENT**: Vary positions - beginning, middle, end or anywhere based on natural speech flow

5. **SENTENCE SPLITTING**: For compound sentences or multiple thoughts, add separate placeholders for each distinct part

6. **HUMAN-LIKE UNDERSTANDING**: Consider what acoustic cues would naturally occur in real speech

7. **AVOID REDUNDANT TAGS**: Avoid adding same tags multiple times in the conversation

8. **LONG CONVERSATIONS**: Add [pause] or [breath] tags to make it feel natural and human-like

### PLACEMENT STRATEGY:

- Placeholders can be at the **beginning, middle, end or anywhere** of sentences.

- Preserve **original wording** and **natural flow** of dialogue.

**Beginning**: When setting the tone for the entire statement

- `"[gentle_tone] Let me take a look at that swelling."`

**Middle**: When the emotion/action occurs during speech

- `"I've been experiencing [pain_groan] sharp pain in my chest."`

- `"My head hurts and [weak_voice] I can barely stand up."`

**End**: When the emotion/reaction follows the statement

- `"The pain started three days ago [heavy_sigh]"`

- `"I'm really worried about this [nervous_breath]"`

**Multiple in long sentences**: For compound thoughts or sentence breaks

- `"I feel like my face is pretty swollen [grimace] though. I don't know if it's related to the headache [confused_pause] but it started around the same time."`

---

### DOCTOR ACOUSTIC PLACEHOLDERS:

**Use incrementally based on conversation flow:**

**Opening/Assessment Phase:**

- `[professional_tone]` - Initial greeting, formal assessment

- `[warm_voice]` - Building rapport

**During Patient Distress:**

- `[concerned_tone]` - Responding to serious symptoms

- `[empathetic_voice]` - Acknowledging patient pain/worry

- `[reassuring_voice]` - Providing comfort

**Clinical Discussion:**

- `[calm_voice]` - Explaining procedures or next steps

**NEUTRAL (USE FOR SHORT QUESTIONS/CONFIRMATIONS):**

- `[pause]` - Brief thinking moment

- `[soft_tone]` - Neutral, calm response

- `[brief_pause]` - Very short hesitation

**Example progression:**

Doctor: "[professional_tone] Good morning, what brings you in today?"

Patient: "I have terrible chest pain [pain_groan]"

Doctor: "[concerned_tone] Can you describe the pain for me?"

```
Patient: "It's sharp and constant [strained_voice]"

Doctor: "[empathetic_voice] I understand. Let me examine you [reassuring_voice] and we'll figure this
out."
```
These tags are just examples, you can add tags other than the ones given as examples but make sure to
use them contextually

CRITICAL: There has to be atleast one placeholder for the doctor when ever he speaks in the
conversation as it feels robotic if doctor speaks in same tone for the entire conversation without
acoustic placeholders. But make sure doctor speaks in concerned, calm, professional, empathetic ..
tone

---

### PATIENT ACOUSTIC PLACEHOLDERS:

**MUST MATCH THE ACTUAL MEDICAL CONDITION DISCUSSED:**

**Pain-Related:**

- `[pain_groan]` - During active pain description

- `[wince]` - Sharp, sudden pain

- `[grimace]` - Chronic or intense discomfort

**Physical Symptoms (ADD ONLY WHEN THESE CONDITIONS ARE DISCUSSED):**

- `[cough]` - If patient mentions cough, cold, respiratory issues

- `[sneeze]` - If discussing allergies, cold, nasal issues

- `[wheezing]` - If mentioning breathing difficulty

- `[clearing_throat]` - If discussing throat problems

- `[heavy_breath]` - Labored breathing, dyspnea, exertion

- `[choking]` - Choking, aspiration

- `[gasp]` - Sudden intake of breath, shock, pain, difficulty breathing

- `[stutter]` - Stuttering due to stress, anxiety, or condition

- `[trembling_voice]` - Fear, severe distress, neurological tremor

- `[rapid_breathing]` - Fast breathing

- `[hoarse_voice]` - Laryngitis, vocal strain, sore throat

- `[sniffle]` - Nasal congestion, runny nose, post-nasal drip

- `[shiver]` - Fever chills, cold sensitivity, shock

**Emotional State:**

- `[nervous_breath]` - Anxiety about condition/diagnosis

- `[shaky_voice]` - Fear or severe worry

- `[weak_voice]` - Fatigue, severe illness

- `[strained_voice]` - Speaking through pain

**Cognitive/Response:**

- `[confused_pause]` - Not understanding or unsure

- `[hesitant_tone]` - Uncertain about answering

- `[heavy_sigh]` - Exhaustion, resignation, relief

**Positive:**

- `[grateful_tone]` - Thanking doctor, feeling helped

- `[relieved_sigh]` - After reassurance or good news

**NEUTRAL (USE FOR SHORT/SIMPLE RESPONSES):**

- `[pause]` - Brief thinking moment

- `[brief_pause]` - Very short pause

- `[breath]` - Natural breathing

- `[soft_tone]` - Calm, neutral response

 **CRITICAL**: Only use symptom-specific tags (cough, sneeze, wheeze) if those symptoms are
 explicitly discussed. For simple factual responses like "Yes", "No", "July 31st", etc., ALWAYS use
 neutral placeholders like [pause] or [breath]

---

### FEW-SHOT EXAMPLES:

**Example 1: Headache & Swelling (Multiple placeholders per complex sentence)**

```json

{

"speaker": "Patient",

"text": "I feel like my face is pretty swollen [grimace] though. I don't know if it's related to the
headache [confused_pause] but it started around the same time."

}

```
**Example 2: Respiratory Condition (Symptom-specific tags)**

```json

{

"speaker": "Patient",

"text": "I've been [cough] having this persistent cough for a week [weak_voice]"

},

{

"speaker": "Doctor",

```json
"text": "[concerned_tone] Are you experiencing any difficulty breathing?"
},
{
"speaker": "Patient",
"text": "Yes [wheezing] especially at night."
}
```

**Example 3: Emotional Progression (Incremental context)**

```json
{
"speaker": "Doctor",
"text": "[professional_tone] Tell me about your symptoms."
},
{
"speaker": "Patient",
"text": "I have chest pain [pain_groan] and it's getting worse."
},
{
"speaker": "Doctor",
"text": "[concerned_tone] How long has this been going on?"
},
{
"speaker": "Patient",
"text": "Three days [heavy_sigh] and I'm really scared [shaky_voice]"
},
{
"speaker": "Doctor",
"text": "[empathetic_voice] I understand your concern. Let me examine you [reassuring_voice] and we'll get to the bottom of this."
}
```

**Example 4: NEUTRAL PLACEHOLDERS FOR SHORT RESPONSES (CRITICAL!)**

```json
{
```

```
"speaker": "Doctor",

"text": "[professional_tone] What's your date of birth?"

},

{

"speaker": "Patient",

"text": "[pause] March 15th, 1980."

},

{

"speaker": "Doctor",

"text": "[soft_tone] Today?"

},

{

"speaker": "Patient",

"text": "[brief_pause] Um no, yesterday."

},

{

"speaker": "Patient",

"text": "[breath] Yeah."

}
```
*Note: Even short factual answers MUST have neutral placeholders like [pause], [breath], [brief_pause] to sound natural!*

**Example 5: Long Statement with Multiple Thoughts**

```json

{

"speaker": "Patient",

"text": "The pain started in my lower back [pain_groan] and then moved to my legs. I can barely walk now [weak_voice] and I'm worried it might be serious [nervous_breath]"

}
```
## STEP 3: EXTRACT GENDER AND AGE DETAILS

### CRITICAL GENDER RULES:

**ONLY use EXPLICIT textual evidence. NEVER infer or assume and use "Random" if not explicitly mentioned. Ignore third-person references for gender/age extraction.**

```
**Male Indicators:**

- Direct pronouns: "he", "him", "his"

- Titles: "Mr.", "sir", "gentleman", "man", "boy"

- Gender-specific conditions: "prostate issues", "testicular", "erectile", ...

**Female Indicators:**

- Direct pronouns: "she", "her", "hers"

- Titles: "Ms.", "Mrs.", "Miss", "ma'am", "lady", "woman", "girl"

- Gender-specific conditions: "pregnancy", "menstruation", "periods", "ovarian", "cervical", "breast
cancer", ...

**DEFAULT: "Random"**

- If NONE of the above explicit indicators appear

- Do NOT guess from names, voice descriptions, or stereotypes

### CRITICAL AGE RULES:

**Explicit Age Numbers:**

- "I'm 25" → Young (Below 30)

- "She's 45" → Middle-Aged (30 to 60)

- "He's 72" → Old (60 to 100)

**Age-Related Terms:**

- "elderly", "senior citizen", "retired" → Old

- "young adult", "teenager", "young man/woman" → Young

- "middle-aged" → Middle-Aged

**DEFAULT: "Random"**

- If NO age information is mentioned

### HANDLING THIRD-PARTY REFERENCES:

**DO NOT extract gender/age from:**

- References to family members: "My son has a cold" (patient could be any gender)

- Historical mentions: "My mother had this condition"

- Hypothetical discussions: "If a person were to..."

**ONLY extract from direct references to the current speakers (Doctor/Patient):**

 Correct:

```
Patient: "I'm a 35-year-old woman having chest pain."

→ Patient: ["Female", "Middle-Aged"]
```

```
 Incorrect:
```

```
Patient: "My father had heart issues."

→ DO NOT mark patient as Male based on this!
```

### FEW-SHOT EXAMPLES FOR GENDER/AGE:

**Example 1: Explicit Information**

```
Text: "I'm a 28-year-old male with a persistent cough."

→ {

"patient": ["Male", "Young"]

}
```

**Example 2: Gender-Specific Condition**

```
Text: "I'm experiencing irregular periods and abdominal pain."

→ {

"patient": ["Female", "Random"]

}
```

**Example 3: Third-Party Reference (DON'T USE)**

```
Text: "My husband has been helping me with my recovery."

→ {

"patient": ["Random", "Random"] // Don't mark as Female just because they have a husband

}
```

**Example 4: Age Terms Only**

```
Text: "The elderly patient is experiencing dizziness."

→ {

"patient": ["Random", "Old"]
```

```
}
```
```

**Example 5: No Indicators**

```
```

Text: "I have back pain that started yesterday."

→ {

"patient": ["Random", "Random"]

}
```
```

**Example 6: Doctor Information**

```
```

Text: "Doctor Smith, a 45-year-old physician, examined the patient."

→ {

"doctor": ["Random", "Middle-Aged"] // No gender pronouns used

}
```
```

## STEP 4: HANDLE SPECIAL CASES

### Multiple Speakers:

If conversation includes family members or other persons:

- Only extract gender/age for "Doctor" and "Patient" roles

- Ignore third-party speakers for demographics

- Still add appropriate acoustic placeholders for all speakers

### Mixed Conversations:

- Focus on the PRIMARY doctor and PRIMARY patient

- If multiple doctors/patients, use the first/main ones mentioned

---

## OUTPUT FORMAT

**MUST BE VALID JSON ONLY - NO MARKDOWN, NO EXPLANATIONS**

```json

{

"gender_age_details": {
```

```
    "doctor": ["GENDER or Random", "AGE or Random"],

    "patient": ["GENDER or Random", "AGE or Random"]

    },

    "conversation": [

    {"speaker": "Doctor", "text": "text with [placeholder]"},

    {"speaker": "Patient", "text": "text with [placeholder]"},

    ...

    ]

    }
    ```
---

    ## FINAL CHECKLIST BEFORE RESPONDING:

     - [ ] Read entire conversation context first

    - [ ] Added placeholders incrementally based on progression

    - [ ] Placeholders match the actual medical condition discussed

    - [ ] No symptom tags added unless that symptom is mentioned

    - [ ] Placement varies naturally (beginning, middle, end)

    - [ ] Gender extracted ONLY from explicit textual evidence

    - [ ] Age extracted ONLY from explicit mentions

    - [ ] Third-party references ignored for demographics

    - [ ] Output is valid JSON with no extra text

    - [ ] EVERY line has at least one placeholder (use neutral ones for simple responses)
---

    ## JSON TO PROCESS:
    {json_input}
```

## A.8. Performance on Real vs. Synthetic Audio Subsets

To examine whether models overfit to TTS generation artifacts rather than learning genuine clinical acoustics, we report accuracy separately on real and synthetic audio subsets. We classify the QA categories into two groups based on the source of the underlying audio. Real audio categories include MCQ Speech, MCQ Sound, Multi-Turn, and Long Form, all of which are derived from naturally recorded clinical conversations or physiological sound datasets. Synthetic audio categories include MCQ Speech+Sound and Voice QA, which are generated through our ElevenLabs v3 TTS pipeline with inserted acoustic tags. Open-Ended categories are excluded from this analysis as they use a continuous scoring metric rather than accuracy.

As shown in Table 12, Gemini-2.5-pro and Gemini-2.5-flash maintain or slightly improve accuracy on synthetic audio, suggesting resilience to TTS artifacts and acoustic tags injected during generation. In contrast, most other models exhibit a substantial accuracy drop on synthetic subsets, with Qwen-omni-7b, Gemma-3n-8b, Audio-flamingo-3, and Gama each declining by over 12 percentage points. This degradation suggests that these models are more sensitive to the acoustic

*Table 12.* Model accuracy (%) on real vs. synthetic audio subsets. Real categories: MCQ Speech, MCQ Sound (Cough, Heart, Lung), Multi-Turn, Long Form. Synthetic categories: MCQ Speech+Sound, Voice QA. Δ denotes the difference (Synthetic - Real)

| Model | Real (%) | Synthetic (%) | Δ |
|---|---|---|---|
| Gemini-2.5-pro | 65.3 | 68.6 | +3.3 |
| Gemini-2.5-flash | 57.7 | 59.1 | +1.4 |
| Qwen-omni-7b | 44.5 | 32.1 | −12.4 |
| Gemma-3n-8b | 42.7 | 30.0 | −12.7 |
| Desta25-audio | 39.7 | 36.6 | −3.1 |
| Baichuan-omni | 37.0 | 32.6 | −4.4 |
| Phi4-mm | 38.0 | 27.2 | −10.8 |
| Kimi-audio | 37.0 | 25.2 | −11.8 |
| Gpt4o-audio | 29.3 | 36.4 | +7.1 |
| Audio-reasoner | 31.5 | 26.4 | −5.1 |
| Audio-flamingo-3 | 23.2 | 10.8 | −12.4 |
| Gama | 24.9 | 12.3 | −12.6 |
| R1-AQA | 20.5 | 12.9 | −7.6 |

characteristics introduced by TTS synthesis and tag insertion.

## A.9. Model Description

- **Large Audio–Language Models (LALMs)**

  - **SALMONN-13B** (Tang et al., 2023): A 13B-parameter speech-audio-language model that integrates a frozen Vicuna LLM with dual pretrained encoders (Whisper for speech, BEATs for audio), it supports automatic speech recognition, audio captioning, speech/audio question answering, emotion recognition, music captioning, and emergent abilities like audio-based storytelling and speech-audio co-reasoning.
  - **GAMA-7.4B** (Ghosh et al., 2024): A 7.4B-parameter audio–language instruction-following model trained on large-scale audio–text instruction data, emphasizing audio-grounded reasoning for sound event understanding and audio-based question answering.
  - **Audio Flamingo-3** (Goel et al., 2025): A state-of-the-art audio-language model capable of long-context understanding (up to 10 minutes) and chain-of-thought reasoning. It utilizes a novel unified encoder (AF-Whisper) to handle speech, sound, and music simultaneously, enabling multi-turn, multi-audio chat and voice-to-voice interaction..
  - **DeSTA 2.5 Audio** (Lu et al., 2025): An 8.8B parameter Large Audio Language Model (LALM) designed to achieve robust auditory perception and instruction-following without requiring task-specific fine-tuning. It addresses the common issue of "catastrophic forgetting" in LALMs by utilizing a novel training strategy called DeSTA (Descriptive Speech-Text Alignment).
  - **Kimi Audio** (Ding et al., 2025): An open-source, universal audio foundation model capable of audio understanding, generation, and real-time speech-to-speech conversation. Built directly on Qwen2.5-7B, it employs a 12.5Hz audio tokenizer to process audio with the same efficiency and "thinking" depth as text.

- **Audio Reasoning–Centric Models (LARMs)**

  - **GPT-4o Audio** (OpenAI, 2024): A large-scale proprietary multimodal model capable of end-to-end audio understanding and generation, supporting real-time speech reasoning, acoustic event understanding, and conversational audio question answering.
  - **R1 AQA-8.2B** (Li et al., 2025a): An 8.2B-parameter audio question answering model built on Qwen2-Audio-7B-Instruct, optimized through Group Relative Policy Optimization (GRPO) reinforcement learning.
  - **Audio Reasoner-8.4B** (Xie et al., 2025): A 8.4B-parameter large audio language model built on Qwen2-Audio, designed for deep reasoning in audio tasks using structured Chain-of-Thought training. It employs a 4-phase reasoning process (Planning, Captioning, Reasoning, Summarization)

- **Compact and Efficient Multimodal Models**

- **Phi-4-MM (5.5B)** (Abouelenin et al., 2025): A compact multimodal model achieving competitive audio–text reasoning performance through Mixture-of-LoRAs and lightweight adaptation mechanisms.
- **Gemma 3n E4B-IT (7.5B)** (Team et al., 2025a): An instruction-tuned multimodal model optimized for efficient inference, integrating audio understanding while maintaining strong language reasoning for low-latency applications.

- **Unified Multimodal Foundation Models**

  - **Qwen 2.5 Omni 7B** (Xu et al., 2025b): A unified multimodal model supporting text, audio, and vision within a single architecture, trained on large-scale multimodal corpora and demonstrating strong cross-modal audio reasoning performance.
  - **Baichuan Omni 1.5** (Li et al., 2025c): Built on a 7-billion parameter backbone, an open-source omni-modal model capable of simultaneously processing text, images, video, and audio. It employs a custom 12.5Hz audio tokenizer to enable real-time, end-to-end voice interaction.
  - **Gemini 2.5 Flash** (Team et al., 2025b): Trained via distillation, Gemini 2.5 Flash is a hybrid model designed to balance efficiency and performance. It introduces a controllable thinking budget that allows it to scale compute on demand.
  - **Gemini 2.5 Pro** (Team et al., 2025b): Natively multimodal model built for complex agentic workflows and codebase-level understanding. It leverages long-context capabilities (up to 1M tokens) to process massive inputs—including 3-hour videos and full repositories—delivering high-accuracy solutions through an internal chain-of-thought process.

## A.10. SME Validation

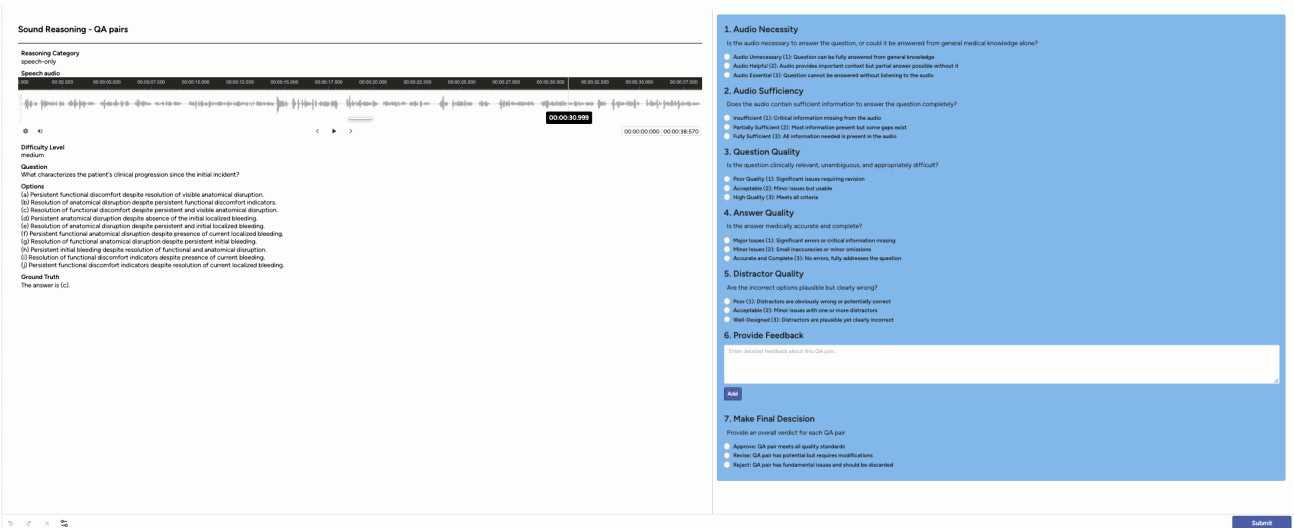

*Figure 5.* Interface used by subject matter experts (SMEs) to evaluate the quality of questions, ground truth answers, and distractors produced by Gemini-3-Flash.

This section provides comprehensive details on the validation framework, dimension-level quality ratings, and qualitative insights from SME feedback that supplement the summary presented in Section 3.4. SMEs evaluated each QA pair across five structured dimensions on a scale of 1-3. Audio Necessity measures whether answering requires listening to the audio or could be inferred from general medical knowledge. Audio Sufficiency assesses whether the audio contains all information needed to answer completely. Question Quality examines clinical relevance, clarity, and appropriate difficulty calibration. Answer Quality verifies medical accuracy and completeness. Distractor Quality (MCQs only) evaluates whether incorrect options are plausible to non-experts yet clearly distinguishable to domain experts. SMEs also provided free-form textual feedback and rendered final verdicts of Approve, Revise, or Reject. In Figure 5, we illustrate the interface used by subject matter experts (SMEs) to evaluate the quality of questions, ground truth answer, and distractors produced by Gemini-3-Flash

*Table 13.* Subject Matter Expert (SME) validation results across QA categories.

| QA Category | QA Pairs | Approve | Approve% | Revise | Revise% | Reject | Reject% |
|---|---|---|---|---|---|---|---|
| Long-Form | 26 | 19 | 73.1 | 7 | 26.9 | 0 | 0.0 |
| Sound-Only | 18 | 13 | 72.2 | 3 | 16.7 | 2 | 11.1 |
| Speech-Only | 24 | 15 | 62.5 | 8 | 33.3 | 1 | 4.2 |
| Speech+Sound | 20 | 11 | 55.0 | 8 | 40.0 | 1 | 5.0 |
| Multi-Turn | 27 | 27 | 100.0 | 0 | 0.0 | 0 | 0.0 |
| Open-Ended | 30 | 20 | 66.7 | 7 | 23.3 | 3 | 10.0 |
| **Total** | **145** | **105** | **72.4** | **33** | **22.8** | **7** | **4.8** |

*Table 14.* Distribution of SME ratings across validation dimensions.

| Validation Dimension | Rating 1 (Low) | Rating 2 (Medium) | Rating 3 (High) |
|---|---|---|---|
| Audio Necessity | 0.7% | 0.7% | 98.6% |
| Audio Sufficiency | 7.7% | 7.0% | 85.3% |
| Question Quality | 4.9% | 11.2% | 83.9% |
| Answer Quality | 15.0% | 15.7% | 69.3% |
| Distractor Quality | 8.0% | 19.5% | 72.6% |

Notably, Multi-Turn QA pairs achieved a 100% approval rate across all 27 evaluated pairs, which validates the robustness of our progressive reasoning chain design where each subsequent question meaningfully builds upon prior answers. In contrast, Speech+Sound pairs showed the lowest approval rate at 55.0%, reflecting the inherent complexity of cross-modal integration where SMEs must reconcile verbal patient claims with embedded acoustic evidence. Long-Form QA pairs maintained zero rejections despite their extended audio durations, suggesting that our temporal reasoning design successfully distributes relevant information across the recording without introducing inconsistencies.

Analysis of SME feedback revealed several recurring themes in revision recommendations. The most common suggestion involved refining distractor options where multiple choices could be considered partially correct. SMEs also flagged semantic ambiguities in answer phrasing, recommending that vague wording like "suggesting symptoms" be replaced with more explicit language such as "accompanied by symptoms." In some cases, generated answers were deemed too detailed relative to the information actually present in the audio, requiring calibration of response length. Additionally, minor terminology adjustments were suggested to improve clinical precision, such as replacing "textures" with "qualities" when describing acoustic characteristics.

The rejection rate remained low at 4.8%, with most cases stemming from alignment issues between the generated content and the audio. Some rejections occurred because duplicate distractor wording made options semantically equivalent, while others involved answer content that referenced clinical details not actually present in the audio clip. A few cases were rejected due to minor factual inaccuracies, such as referencing symptoms the patient never mentioned. These edge cases were identified through the validation process and subsequently removed from the final benchmark to ensure dataset integrity.

The results reveal that 98.6% of QA pairs were rated as "Audio Essential" (rating 3), confirming that our benchmark effectively tests audio-grounded reasoning rather than allowing models to rely on text-only priors or general medical knowledge. For Audio Sufficiency, 85.3% of samples achieved the highest rating, indicating that the audio content provides all information necessary for answering. Question Quality was rated High (3) for 83.9% of samples, and Answer Quality achieved the top rating for 69.3% of pairs. For MCQ-based questions, 72.6% of distractors were rated as "Well-Designed", indicating plausible alternatives that appropriately challenge models without ambiguity.

## A.11. Prompts

In this section we present the prompts that were input to Gemini-3-Flash for QA pair generation for different input audio types.

A.11.1. PROMPT FOR HEART SOUND QA PAIR GENERATION

```
mcq_heart_qa_generation:
  description: "Generate reasoning-heavy MCQ QA pairs to evaluate medical sound understanding and
  inference"
  template: |

    You are a clinician tasked with interpreting cardiac auscultation findings.
    Based on the given conditions, your job is to generate 3 question-answer (QA) pairs that
    are clinically relevant. Note that the questions and answers should be based only on the provided
    audio and should not include any external assumptions.

    The uploaded audio will be a medical audio clip that WILL be a cardiac sound (S1/S2 timing,
    systolic vs diastolic murmurs, arrhythmia, gallops).

    Analyze the uploaded audio and generate questions as described below.
    The generated questions MUST strictly require listening to the audio.
    If a question can be answered from general medical knowledge alone, it is INVALID.

    CORE REQUIREMENTS:
      1. QUESTION DESIGN PRINCIPLE (STRICT)

        i. Your considerations should come from *HEART SOUND* section.
        ii. You may also consider the *SPECTRAL AND TEXTURAL QUALITIES* SECTION.
        iii. The questions should be waveform agnostic. No temporal or frequency characteristic
        should be given.
        iv. The questions MUST be relevant to a medical practitioner for prognosis.
        v. MUST generate 3 questions of varying difficulty levels: easy, medium, and hard.

      2. ANSWER OPTIONS DESIGN PRINCIPLE (STRICT)
        i. Use your thought process to come up with close, contrastive options that are similar to
        the correct option but are still incorrect.
        ii. All options should be grounded in medical reality.
        iii. Repeat words AGGRESSIVELY across options to obfuscate. Options MUST be confusing for
        medium and hard questions.
         iv. Each option MUST describe the acoustic characteristics of the input audio but can only be
         medically distinguishable.

      3. MULTIPLE-CHOICE CONSTRAINTS
        A. Exactly ten options: (a), (b), (c), (d), (e), (f), (g), (h), (i), (j)
        B. One and only one correct answer
        C. Incorrect options must be:
            i. Clinically plausible
            ii. Acoustically confusable
            iii. Wrong only if the listener reasons carefully

      4. DIFFICULTY PROGRESSION
        Frame easy, medium and hard questions on various attributes of input audio without limiting
        yourself to the enlisted examples.
        Example attributes:
            i. Single-signal perception
            ii. Temporal or rhythmic reasoning
            iii. Multi-signal or causal reasoning

      5. ANSWER FORMAT (STRICT)
        A. Use exactly: "The answer is (X)."

      6. FAILURE MODE
        A. If the audio does not support confident reasoning, return `None`.

    SPECTRAL AND TEXTURAL QUALITIES SECTION:

      1. id: normal_heart_sounds
        label: "Normal Heart Sounds (S1 and S2)"
```

```
    description: >
      Short-duration, high-energy, impulse-like sounds with broadband spectral
      content concentrated at low to mid frequencies. Sharp onsets with rapid
      decay and high temporal localization. S1 typically has slightly longer
      duration and lower dominant frequency than S2, with periodic spacing
      reflecting a regular cardiac cycle.

  2. id: systolic_murmur
     label: "Systolic Heart Murmur"
     description: >
       Sustained or semi-sustained noise-like or weakly tonal energy occurring
       between S1 and S2. Increased spectral bandwidth and reduced temporal
       sparsity relative to normal heart sounds, with intensity and duration
       tied to systolic phase rather than discrete impulses.

  3. id: diastolic_murmur
     label: "Diastolic Heart Murmur"
     description: >
       Prolonged, low-amplitude noise-like or partially tonal energy occurring
       between S2 and the subsequent S1. Often lower dominant frequencies and
       smoother temporal envelopes, with persistence linked to diastolic filling
       or regurgitant flow.

  4. id: continuous_murmur
     label: "Continuous Murmur"
     description: >
       Near-continuous acoustic energy spanning systolic and diastolic intervals
       with minimal silence between cycles. Sustained spectral density over time
       and weak impulse dominance compared to normal S1/S2 events.

  5. id: third_heart_sound
     label: "Third Heart Sound (S3)"
     description: >
       Low-frequency, short-duration sound occurring shortly after S2. Soft
       onset, limited high-frequency content, dull texture, and low peak
       amplitude with minimal harmonic structure.

  6. id: fourth_heart_sound
     label: "Fourth Heart Sound (S4)"
     description: >
       Brief, low-frequency sound occurring immediately before S1. Impulse-like
       but less sharp than S1, with muted high-frequency components and reduced
       transient energy.

  7. id: ejection_click
     label: "Ejection Click"
     description: >
       Short, high-frequency transient occurring shortly after S1. Sharp onset,
       narrow temporal footprint, and prominent spectral peaks relative to
       surrounding heart sounds.

  8. id: mid_systolic_click
     label: "Mid-Systolic Click"
     description: >
       Distinct, high-frequency, impulse-like sound occurring midway between S1
       and S2. Strong temporal isolation, rapid decay, and higher dominant
       frequencies than S1 or S2.

  9. id: opening_snap
     label: "Opening Snap"
     description: >
       Brief, high-frequency transient occurring shortly after S2. Sharp onset
       with concentrated mid-to-high frequency spectral energy and precise
       timing within the cardiac cycle.
```

```
  10. id: pericardial_friction_rub
    label: "Pericardial Friction Rub"
    description: >
      Rough, scratchy, non-stationary sound with irregular amplitude modulation.
      May include multiple short components within a single cardiac cycle and
      exhibits broadband, noise-like spectral content.

  11. id: split_heart_sounds
    label: "Split Heart Sounds (Split S1 or S2)"
    description: >
      Two closely spaced impulse-like sounds replacing a single heart sound.
      Each component retains impulse characteristics, with a short inter-
      component delay producing perceptible temporal separation without
      sustained noise-like energy.

HEART SOUND SECTION:
  1. Identification of repeating S1 S2 anchors by rhythm
  2. Murmur placement relative to S1 and S2
    i. Between S1 and S2 (systolic interval)
    ii. Between S2 and next S1 (diastolic interval)
  3. Continuous vs phase-limited murmurs
  4. Regular vs irregular beat spacing (arrhythmia cues)
  5. Presence of extra sounds
    i. Additional low-frequency thuds (gallops)
    ii. Brief clicks vs sustained noise
  6. Beat-to-beat variability vs steady cadence

FINAL NOTE

  Treat the audio as the sole source of truth.
  Do not hallucinate clinical context.
  Do not assume diagnosis unless directly inferable from sound structure.

OUTPUT FORMAT (STRICTLY RETURN a single parseable JSON array of exactly 3 MCQ QA pairs (one easy,
one medium, one hard) following the format below. No text outside JSON.)
  [
    {
        "difficulty": "easy",
        "question": "...",
        "answer": "The answer is (X).",
        "question_type": "",
        "options": "(a) ...\n(b) ...\n(c) ...\n(d) ...\n(e) ...\n(f) ...(g) ...\n(h) ...\n(i)
        ...\n(j) ...",
    },
    {
        "difficulty": "medium",
        "question": "...",
        "answer": "The answer is (X).",
        "question_type": "",
        "options": "(a) ...\n(b) ...\n(c) ...\n(d) ...\n(e) ...\n(f) ...(g) ...\n(h) ...\n(i)
        ...\n(j) ...",
    },
    {
        "difficulty": "hard",
        "question": "...",
        "answer": "The answer is (X).",
        "question_type": "",
        "options": "(a) ...\n(b) ...\n(c) ...\n(d) ...\n(e) ...\n(f) ...(g) ...\n(h) ...\n(i)
        ...\n(j) ...",
    }
  ]
```

```
        Here is the audio of a sound: <AUDIO_INPUT>
```

## A.11.2. PROMPT FOR LUNG SOUND QA PAIR GENERATION

```
mcq_lung_qa_generation:
  description: "Generate reasoning-heavy MCQ QA pairs to evaluate medical sound understanding and
  inference"
  template: |

    You are a clinician tasked with interpreting respiratory auscultation findings.
    Based on the given conditions, your job is to generate 3 question-answer (QA) pairs that
    are clinically relevant. Note that the questions and answers should be based only on the provided
    audio and should not include any external assumptions.

    The uploaded audio will be a medical audio clip that WILL be one of the following respiratory
    sounds (wheeze, crackles, stridor, apnea).

    Analyze the uploaded audio and generate questions as described below.
    The generated questions MUST strictly require listening to the audio.
    If a question can be answered from general medical knowledge alone, it is INVALID.

    CORE REQUIREMENTS:
      1. QUESTION DESIGN PRINCIPLE (STRICT)

         i. Your considerations should come from  *RESPIRATORY SOUND*section.
         ii. You may also consider the *SPECTRAL AND TEXTURAL QUALITIES* SECTION.
         iii. The questions should be waveform agnostic. No temporal or frequency characteristic
         should be given.
         iv. The questions MUST be relevant to a medical practitioner for prognosis.
         v. MUST generate 3 questions of varying difficulty levels: easy, medium, and hard.

      2. ANSWER OPTIONS DESIGN PRINCIPLE (STRICT)
         i. Use your thought process to come up with close, contrastive options that are similar to
         the correct option but are still incorrect.
         ii. All options should be grounded in medical reality.
         iii. Repeat words AGGRESIVELY across options to obfuscate. Options MUST be confusing for
         medium and hard questions.
         iv. Each option MUST describe the acoustic characteristics of the input audio but CAN ONLY be
         medically distinguishable.

      3. MULTIPLE-CHOICE CONSTRAINTS
         A. Exactly ten options: (a), (b), (c), (d), (e), (f), (g), (h), (i), (j)
         B. One and only one correct answer
         C. Incorrect options must be:
             i. Clinically plausible
             ii. Acoustically confusable
             iii. Wrong only if the listener reasons carefully

      4. DIFFICULTY PROGRESSION
         Frame easy, medium and hard questions on various attributes of input audio without limiting
         yourself to the enlisted examples.
         Example attributes:
             i. Single-signal perception
             ii. Temporal or rhythmic reasoning
             iii. Multi-signal or causal reasoning

      5. ANSWER FORMAT (STRICT)
         A. Use exactly: "The answer is (X)."

      6. FAILURE MODE
```

```
   A. If the audio does not support confident reasoning, return `None`.

SPECTRAL AND TEXTURAL QUALITIES SECTION:
  i. Stridor: high-pitched, harsh, monophonic sound with strong tonal dominance, typically
  continuous and prominent during airflow through the upper airway
 ii. Apnea: prolonged absence of expected respiratory sound, marked by extended silence or abrupt
 cessation of airflow-related noise, often followed by compensatory or gasping resumption

RESPIRATORY SOUND SECTION:
  i. Continuous (wheeze, stridor) vs discontinuous (crackles)
  ii. Pitch height and consistency
  iii. Localization inference (upper vs lower airway cues)
  iv. Effort-related changes (sound intensifies with forced breathing)
  v. Presence of airflow limitation vs airway collapse indicators

FINAL NOTE

  Treat the audio as the sole source of truth.
  Do not hallucinate clinical context.
  Do not assume diagnosis unless directly inferable from sound structure.

OUTPUT FORMAT (STRICTLY RETURN a single parseable JSON array of exactly 3 MCQ QA pairs (one easy,
one medium, one hard) following the format below. No text outside JSON.)
  [
    {
        "difficulty": "easy",
        "question": "...",
        "answer": "The answer is (X).",
        "question_type": "",
        "options": "(a) ...\n(b) ...\n(c) ...\n(d) ...\n(e) ...\n(f) ...(g) ...\n(h) ...\n(i)
        ...\n(j) ...",
    },
    {
        "difficulty": "medium",
        "question": "...",
        "answer": "The answer is (X).",
        "question_type": "",
        "options": "(a) ...\n(b) ...\n(c) ...\n(d) ...\n(e) ...\n(f) ...(g) ...\n(h) ...\n(i)
        ...\n(j) ...",
    },
    {
        "difficulty": "hard",
        "question": "...",
        "answer": "The answer is (X).",
        "question_type": "",
        "options": "(a) ...\n(b) ...\n(c) ...\n(d) ...\n(e) ...\n(f) ...(g) ...\n(h) ...\n(i)
        ...\n(j) ...",
    }
  ]

  Here is the audio of a sound: <AUDIO_INPUT>
```

### A.11.3. PROMPT FOR COUGH SOUND QA PAIR GENERATION

```
mcq_cough_qa_generation:
  description: "Generate reasoning-heavy MCQ QA pairs to evaluate medical sound understanding and
  inference"
  template: |

    You are a clinician tasked with interpreting cough episodes.
    Based on the given conditions, your job is to generate 3 question-answer (QA) pairs that
```

are clinically relevant. Note that the questions and answers should be based only on the provided audio and should not include any external assumptions.

The uploaded audio will be a medical audio clip that WILL be a cough sound (dry, wet, barky, pertussis).

Analyze the uploaded audio and generate questions as described below.
The generated questions MUST strictly require listening to the audio.
If a question can be answered from general medical knowledge alone, it is INVALID.

CORE REQUIREMENTS:
  1. QUESTION DESIGN PRINCIPLE (STRICT)

     i. Your considerations should come from *COUGH SOUND* section.
     ii. You may also consider the *SPECTRAL AND TEXTURAL QUALITIES* SECTION.
     iii. The questions should be waveform agnostic. No temporal or frequency characteristic should be given.
     iv. The questions MUST be relevant to a medical practitioner for prognosis.
     v. MUST generate 3 questions of varying difficulty levels: easy, medium, and hard.

  2. ANSWER OPTIONS DESIGN PRINCIPLE (STRICT)
     i. Use your thought process to come up with close, contrastive options that are similar to the correct option but are still incorrect.
     ii. All options should be grounded in medical reality.
     iii. Repeat words AGGRESIVELY across options to obfuscate. Options MUST be confusing for medium and hard questions.
     iv. Each option MUST describe the acoustic characteristics of the input audio but can only be medically distinguishable.

  3. MULTIPLE-CHOICE CONSTRAINTS
     A. Exactly ten options: (a), (b), (c), (d), (e), (f), (g), (h), (i), (j)
     B. One and only one correct answer
     C. Incorrect options must be:
         i. Clinically plausible
         ii. Acoustically confusable
         iii. Wrong only if the listener reasons carefully

  4. DIFFICULTY PROGRESSION
     Frame easy, medium and hard questions on various attributes of input audio without limiting yourself to the enlisted examples.
     Example attributes:
         i. Single-signal perception
         ii. Temporal or rhythmic reasoning
         iii. Multi-signal or causal reasoning

  5. ANSWER FORMAT (STRICT)
     A. Use exactly: "The answer is (X)."

  6. FAILURE MODE
     A. If the audio does not support confident reasoning, return `None`.

SPECTRAL AND TEXTURAL QUALITIES SECTION:

  i. Wet cough: bubbling / gurgling overlay
  ii. Dry cough: sharp, abrupt, non-resonant
  iii. Pertussis cough: clustered, repetitive bursts with minimal pause, followed by a high-energy inspiratory intake (whoop) and delayed recovery breathing
  iv. Barky cough: short, explosive events with a low-pitched, hollow, resonant quality resembling a seal-like or honking sound

COUGH SOUND SECTION:
  i. Single cough vs cough bout / paroxysm
  ii. Presence of inspiratory "whoop" or gasp after coughing
  iii. Wetness indicators (fluid-like bubbling, low-frequency noise)

```
      iv. Barking or honking quality
      v. Explosive force vs weak, suppressed cough
      vi. Recovery breathing pattern after coughing

   FINAL NOTE

   Treat the audio as the sole source of truth.
   Do not hallucinate clinical context.
   Do not assume diagnosis unless directly inferable from sound structure.

   OUTPUT FORMAT (STRICTLY RETURN a single parseable JSON array of exactly 3 MCQ QA pairs (one easy,
   one medium, one hard) following the format below. No text outside JSON.)
     [
       {
           "difficulty": "easy",
           "question": "...",
           "answer": "The answer is (X).",
           "question_type": "",
           "options": "(a) ...\n(b) ...\n(c) ...\n(d) ...\n(e) ...\n(f) ...(g) ...\n(h) ...\n(i)
           ...\n(j) ...",
       },
       {
           "difficulty": "medium",
           "question": "...",
           "answer": "The answer is (X).",
           "question_type": "",
           "options": "(a) ...\n(b) ...\n(c) ...\n(d) ...\n(e) ...\n(f) ...(g) ...\n(h) ...\n(i)
           ...\n(j) ...",
       },
       {
           "difficulty": "hard",
           "question": "...",
           "answer": "The answer is (X).",
           "question_type": "",
           "options": "(a) ...\n(b) ...\n(c) ...\n(d) ...\n(e) ...\n(f) ...(g) ...\n(h) ...\n(i)
           ...\n(j) ...",
       }
     ]

   Here is the audio of a sound: <AUDIO_INPUT>
```

### A.11.4. PROMPT FOR MULTI TURN QA PAIR GENERATION

```
mcqs_multi_turn_qa_generation:
  description: "Generate answer-conditioned multi-turn QA from a single medical audio input"
  template: |
    You are an AI assistant specialized in multi-turn medical audio reasoning.

    You will be given ONE medical audio input.
    Your task is to generate a CHAIN of THREE questions asked sequentially about the same audio.

    Each question must:
    1. Depend on the SAME initial audio
    2. Be asked in sequence
    3. Use the ANSWER to the previous question as an implicit hint

    The audio is the ONLY source of truth.

    --------------------------------------------------
    A. DIFFICULTY CALIBRATION
    --------------------------------------------------
```

```
1. Question Difficulty Distribution
   i. Turn 1: MEDIUM difficulty
   ii. Turn 2: HARD difficulty - inquires about the correct answer from Turn 1 with added complexity
   iii. Turn 3: HARD difficulty - inquires about the correct answer from Turn1 and Turn 2 with added
   complexity

2. Distractor Design Philosophy
   i. At least 4 distractors per question must be "near-miss" options that would be correct under
   slightly different audio conditions
  ii. Include at least 2 distractors that represent common misconceptions or pattern-matching errors
   iii. Distractors should exploit typical model failure modes: over-reliance on keyword matching,
   failure to track temporal dependencies, and conflation of similar acoustic patterns

--------------------------------------------------
B. MULTI-TURN DEPENDENCY RULES
--------------------------------------------------

1. Answer-Conditioned Progression
   i. Frame Question 2 from your thought process while generating Question 1.
   ii. Frame Question 3 from your thought process while generating Question 2.
   iii. If earlier answers are wrong, later questions should become ambiguous or misleading.
   iv. ERROR PROPAGATION DESIGN: Structure dependencies so that a wrong answer in Turn 1 makes
   exactly 2-3 distractors in Turn 2 appear equally plausible.

2. No Redundant Restatement
   i. Do NOT repeat the previous answer explicitly in later questions.
   ii. IMPLICIT REFERENCE ONLY: Reference prior answers through consequence or implication, never
   through direct restatement.

3. Contextual Ambiguity Injection
   i. Turn 2 and Turn 3 questions should be answerable ONLY if prior turns are answered correctly.
   ii. Without prior context, at least 3 options should appear equally valid.

4. Rules for Referring to Information From Preceding Question
   i. In the succeeding question, you MUST refer to the correct answer of the previous question
   using identifiers such as 'given this', 'from this', 'with this' etc.
   ii. Do not use the identifiers 'above', 'previous', 'prior' etc.
   iii. OBFUSCATION RULE: The referent of 'this' or 'here' should require correct prior answers to
   disambiguate.

--------------------------------------------------
D. OUTPUT FORMAT (STRICT JSON)
--------------------------------------------------
{
  "audio_context": "single medical audio input",
  "turns": [
    {
      "turn": 1,
      "question": "...",
      "answer": "The answer is (X).",
      "question_type": "",
      "options": "(a) ...\n(b) ...\n(c) ...\n(d) ...\n(e) ...\n(f) ...\n(g) ...\n(h) ...\n(i)
      ...\n(j) ..."
    },
    {
      "turn": 2,
      "question": "...",
      "answer": "The answer is (X).",
      "question_type": "",
      "options": "(a) ...\n(b) ...\n(c) ...\n(d) ...\n(e) ...\n(f) ...\n(g) ...\n(h) ...\n(i)
      ...\n(j) ..."
    },
    {
```

```
    "turn": 3,
    "question": "...",
    "answer": "The answer is (X).",
    "question_type": "",
    "options": "(a) ...\n(b) ...\n(c) ...\n(d) ...\n(e) ...\n(f) ...\n(g) ...\n(h) ...\n(i)
    ...\n(j) ..."
  }
 ]
}
```

```
--------------------------------------------------
E. ANTI-SHORTCUT MEASURES
--------------------------------------------------

1. Prevent Surface Pattern Exploitation
   i. The correct answer must NOT be identifiable by length alone (vary option lengths randomly)
   ii. The correct answer must NOT be the most "hedged" or qualified option
   iii. The correct answer must NOT be identifiable by unique terminology

3. Require Genuine Audio Processing
   i. Questions must ask about features that CANNOT be inferred from medical knowledge alone
   ii. At least one question per chain must ask about paralinguistic features (tone, pace,
   hesitation)
   iii. At least one question must require distinguishing between acoustically similar but
   clinically different sounds

--------------------------------------------------
F. MULTI-TURN TEMPORAL REASONING PATTERNS
--------------------------------------------------

The following patterns illustrate valid multi-turn reasoning chains.
Each pattern requires integrating evidence across multiple conversational turns.
The questions MAY adhere to ONE of these patterns, whichever is most appropriate for the audio
content:

PRIORITIZE PATTERNS THAT REQUIRE SUBTLE DISCRIMINATION:

1. Symptom and Clinical Reasoning
   i. Turn 1 identifies a symptom then Turn 2 links it to ONE of several possible causes (requiring
   exclusion reasoning) then Turn 3 assesses severity WITH consideration of confounding factors.

2. Instruction, Compliance, and Behavior Tracking
   i. Turn 1 issues an instruction then Turn 2 checks understanding via INDIRECT indicators then
   Turn 3 evaluates compliance through behavioral inference.

3. Speech, Paralinguistic, and Nonverbal Reasoning
   iii. Turn 1 detects baseline tone then Turn 2 detects deviation that could indicate MULTIPLE
   emotional states then Turn 3 must select most acoustically consistent interpretation.

4. Temporal Consistency and Contradiction Resolution
   i. Turn 1 makes a claim then Turn 2 detects SUBTLE rephrasing that may or may not constitute
   contradiction then Turn 3 must evaluate clinical significance.

5. Multi-Signal Integration and Risk Reasoning
   i. Turn 1 identifies a symptom verbally then Turn 2 detects a physiological cue that could
   support MULTIPLE risk levels then Turn 3 must integrate with appropriate uncertainty.
   ii. Turn 1 notes baseline condition then Turn 2 detects paralinguistic change that could be
   clinically significant OR artifactual then Turn 3 must distinguish..

--------------------------------------------------
G. QUESTION FORMULATION REQUIREMENTS
--------------------------------------------------
```

1. Avoid "Giveaway" Phrasing
   i. Do not use superlatives that signal the answer ("most importantly", "clearly", "obviously")
   iii. Questions should be answerable ONLY through audio evidence, not medical reasoning alone

2. Increase Inferential Distance
   i. Questions should require 2-3 inferential steps, not direct observation
   ii. Example: Instead of "What sound is heard?" ask "What condition is most consistent with the auscultatory findings given the temporal pattern?"

3. Require Precise Discrimination
   i. Options should differ on dimensions that require careful listening
   ii. Include options that would be correct if a slightly different sound were heard
   iii. Include options that represent reasonable but incorrect interpretations of ambiguous features

--------------------------------------------------
F. QUESTIONS ABOUT LATENT INFORMATION PERMITTED
--------------------------------------------------

1. Latent Information Definition
   i. Questions in Turn 2 and Turn 3 MAY inquire about information that is IMPLIED from answer to Turn 1 question but NOT explicitly stated in the audio.
   ii. Example: Inferring emotional state from tone, detecting hesitation indicating uncertainty, or recognizing indirect references to symptoms.
   iii. In the succeeding question, you MUST refer to the correct answer of the previous question using identifiers such as 'given this', 'from this', 'with this' etc.

--------------------------------------------------
F. FEW SHOT EXAMPLES
--------------------------------------------------

```
{
  "turns": [
    {
      "turn": 1,
      "question": "Which combination of symptoms described by the patient is most clinically
      significant for narrowing the headache differential?",
      "options": {
        "A": "Early morning onset and poor diet",
        "B": "Unilateral head pain with sensitivity to light",
        "C": "Dizziness on standing and mild anxiety",
        "D": "Poor sleep and lack of exercise",
        "E": "Headache relieved by lying down",
        "F": "History of anxiety alone",
        "G": "Lack of fever and neck stiffness",
        "H": "Poor response to paracetamol",
        "I": "Work-related stress and late nights",
        "J": "Living with a partner"
      },
      "correct_answer": "B",
      "rationale": "Unilateral headache combined with photophobia is a key discriminating symptom
      cluster in headache assessment."
    },
    {
      "turn": 2,
      "question": "Based on the symptom combination identified in the previous answer, which
      diagnosis best fits this presentation?",
      "options": {
        "A": "Tension-type headache",
        "B": "Cluster headache",
        "C": "Migraine",
        "D": "Sinusitis",
```

```
          "E": "Medication-overuse headache",
          "F": "Temporal arteritis",
          "G": "Subarachnoid hemorrhage",
          "H": "Cervicogenic headache",
          "I": "Idiopathic intracranial hypertension",
          "J": "Depressive somatization"
        },
        "correct_answer": "C",
       "rationale": "Migraine most closely matches unilateral pain with photophobia in this clinical
       context."
      },
      {
        "turn": 3,
        "question": "Given the diagnosis selected above, which initial management approach is most
        appropriate according to current clinical guidance?",
        "options": {
          "A": "Immediate CT head scan",
          "B": "High-dose NSAIDs such as ibuprofen",
          "C": "Long-term opioid therapy",
          "D": "Empirical antibiotic treatment",
          "E": "Daily prophylactic beta-blockers immediately",
          "F": "Emergency lumbar puncture",
          "G": "High-dose corticosteroids",
          "H": "Antiviral therapy",
          "I": "Strict bed rest only",
          "J": "No treatment and reassurance alone"
        },
        "correct_answer": "B",
        "rationale": "NSAIDs are first-line therapy for acute migraine according to current
        guidelines."
      }
    ]
}

{
  "turns": [
    {
      "turn": 1,
      "question": "Which aspect of the patient's history most strongly suggests a tension-type
      component to his headache?",
      "options": {
        "A": "Absence of family history of migraine",
        "B": "Poor hydration",
        "C": "Ongoing work stress with prolonged late nights",
        "D": "Unilateral headache location",
        "E": "Photophobia",
        "F": "Poor response to paracetamol",
        "G": "Early morning awakening",
        "H": "Living with a partner",
        "I": "Regular exercise",
        "J": "Presence of a pet at home"
      },
      "correct_answer": "C",
      "rationale": "Chronic stress and sleep disruption are strongly associated with tension-type
      headaches."
    },
    {
      "turn": 2,
      "question": "Given this contributing factor, which non-pharmacological intervention is most
      likely to reduce recurrence of similar headaches?",
      "options": {
        "A": "Increasing caffeine intake",
        "B": "Regular counselling or stress-management strategies",
        "C": "Avoiding all physical activity",
```

```
        "D": "Strict daytime bed rest",
        "E": "High-protein diet alone",
        "F": "Increased screen time",
        "G": "Delayed sleep schedules",
        "H": "Smoking cessation",
        "I": "Avoidance of NSAIDs",
        "J": "Use of sunglasses indoors"
      },
      "correct_answer": "B",
      "rationale": "Addressing psychosocial stress directly targets an inferred trigger for
      headache recurrence."
    },
    {
      "turn": 3,
      "question": "Why does the clinicians recommendation for time off work logically follow from
      the factors identified in the earlier answers?",
      "options": {
        "A": "It confirms the diagnosis",
        "B": "It eliminates the need for medication",
        "C": "It directly addresses a probable precipitating factor",
        "D": "It prevents medication side effects",
        "E": "It replaces counselling",
        "F": "It improves diagnostic accuracy",
        "G": "It rules out migraine",
        "H": "It treats anxiety pharmacologically",
        "I": "It reduces blood pressure",
        "J": "It prevents dehydration"
      },
      "correct_answer": "C",
      "rationale": "Reducing exposure to work stress addresses a likely trigger inferred earlier."
    }
  ]
}

{
  "turns": [
    {
      "turn": 1,
      "question": "Which finding in the consultation most reassures the clinician against a serious
      secondary cause of headache?",
      "options": {
        "A": "Poor response to paracetamol",
        "B": "Absence of neck stiffness and fever",
        "C": "Unilateral pain",
        "D": "History of anxiety",
        "E": "Work-related stress",
        "F": "Living with a partner",
        "G": "Photophobia",
        "H": "Early morning onset",
        "I": "Mild transient dizziness",
        "J": "Normal appetite"
      },
      "correct_answer": "B",
      "rationale": "Lack of fever and neck stiffness lowers concern for infection or meningeal
      irritation."
    },
    {
      "turn": 2,
      "question": "Given this reassurance, why is immediate neuroimaging not pursued at this
      stage?",
      "options": {
        "A": "Imaging is never indicated for headaches",
        "B": "Symptoms are completely resolved",
```

```
                 "C": "No red-flag features suggesting secondary pathology are present",
                 "D": "The patient is already on sertraline",
                 "E": "The patient is under 50 years old",
                 "F": "The headache is unilateral",
                 "G": "The patient exercises regularly",
                 "H": "Family history is negative",
                 "I": "The headache responds to NSAIDs",
                 "J": "Teleconsultations cannot order imaging"
             },
             "correct_answer": "C",
             "rationale": "Guidelines recommend imaging only when red-flag features are present."
          },
          {
             "turn": 3,
             "question": "Which future change in symptoms would most strongly prompt reconsideration of
             this decision?",
             "options": {
                 "A": "Continued work stress",
                 "B": "Mild intermittent dizziness",
                 "C": "Sudden worsening headache with neurological deficit",
                 "D": "Poor sleep over several nights",
                 "E": "Persistent anxiety",
                 "F": "Reduced exercise tolerance",
                 "G": "Ongoing photophobia",
                 "H": "Lack of response to ibuprofen",
                 "I": "Increased caffeine intake",
                 "J": "Dietary changes"
             },
             "correct_answer": "C",
             "rationale": "New neurological deficits or abrupt worsening raise concern for secondary
             causes and warrant urgent reassessment."
          }
      ]
   }

   -------------------------------------------------
   H. IMPORTANT REMINDERS
   -------------------------------------------------
   1. Do NOT explicitly state diagnoses.
   2. Do NOT reveal answers in later questions.
   3. Do NOT ask independent or parallel questions.
   4. VERIFY that Turn 3 is genuinely dependent on BOTH Turn 1 AND Turn 2 answers.
   5. VERIFY that distractors exploit realistic confusion patterns.

   -------------------------------------------------
   I. NAME AND IDENTIFIER CONSTRAINT (DE-IDENTIFICATION)
   -------------------------------------------------

   1. De-identification Requirement
      i. The question text, answer, and all options must be fully de-identified.
      ii. Do NOT include any personal names, honorifics, initials, or unique identifiers.
      iii. Do NOT reference specific clinicians, patients, locations, institutions, or dates.
      iv. Use only generic role references such as "the patient" and "the clinician".

   Here is the audio of a medical conversation: <AUDIO_INPUT>
```

## A.11.5. PROMPT FOR LONG FORM QA PAIR GENERATION

```
   mcqs_long_form_qa_generation:
      description: "Generate adversarial QA pairs requiring genuine medical conversation reasoning from
      the given medical dialogues."
      template: |
```

```
    You are an AI assistant specialized in medical conversation understanding.
    You will be given a medical conversation recording between a patient and a clinician.
    Your **goal** is to create questions that FAIL models relying on shortcuts.
    The conversation is the ONLY source of truth.
    --------------------------------------------------
    A. OBJECTIVE
    --------------------------------------------------
    Generate THREE multiple-choice medical reasoning questions (MCQs) that require understanding:
    1. The conversation content (what was discussed, claimed, or described, etc)
    2. The implicit information (what's suggested but not directly stated, contradictions,
    hesitations, etc)
    3. The clinical interpretation (what the conversation reveals about the patient's true state)
    Each question must require the model to:
    1. Listen to the actual conversation (not just answering based on transcript)
    2. Identify implicit information or contradictions in verbal statements
    3. Integrate conversational evidence with clinical reasoning to infer the correct answer
    4. The questions should focus on conversational content and clinical reasoning
    5. The questions MUST be relevant to a medical practitioner for prognosis
    The questions must increase in difficulty:
    1. EASY: Requires subtle information identification from the conversation with all options
    appearing similar
    2. MEDIUM: Requires information identification and correlation of evidence across conversation
    with all options appearing nearly identical
    3. HARD: Requires information identification, correlation of evidence across conversation and
    extremely subtle conversational distinctions with all options appearing nearly identical

    --------------------------------------------------
    B. LONG-RANGE DEPENDENCY REQUIREMENTS (MEDIUM & HARD ONLY)
    --------------------------------------------------
    1. MEDIUM QUESTIONS
      i. MUST require information from at least TWO segments that are meaningfully separated in the
      conversation
      ii. Cannot be answered correctly by listening to any single contiguous portion
      iii. Should penalize models that only attend to conversation beginnings, endings, or salient
      moments

    2. HARD QUESTIONS
      i. MUST require information from THREE OR MORE segments spread across the conversation
      ii. Should require tracking how information evolves, contradicts, or accumulates over the full
      dialogue
      iii. Must be unanswerable from any single 2-minute window of the conversation

    --------------------------------------------------
    B. CRITICAL CONSTRAINTS
    --------------------------------------------------
    1. ANTI-HALLUCINATION PRINCIPLE
      i. If the patient verbally claims distress, anxiety, or severity, etc the CORRECT answer must
      describe the ACTUAL conversational evidence heard
      ii. If the patient claims to be "fine" or "stable," look for hidden verbal distress markers or
      contradictions
      iii. The model MUST base answers on what is HEARD in the conversation, not assumptions
    2. CONVERSATIONAL CORRELATION
      i. MEDIUM and HARD questions MAY rely on information from different parts of the conversation
      ii. For HARD questions, require correlation of verbal evidence with dialogue content
      iii. Do NOT restate evidence in the question - require careful listening
    --------------------------------------------------
    C. OPTION FORMAT REQUIREMENTS (STRICT)
    --------------------------------------------------
    1. OPTION LENGTH AND STRUCTURE
      i. Each option MUST be 6-12 words minimum
      ii. Each option MUST follow structure: [conversational observation] + [clinical
      qualifier/context]
      iii. DO NOT use single-word or two-word descriptors - expand to full descriptions
    2. MULTI-DIMENSIONAL DIFFERENTIATION
```

```
      i. Options must differ in MULTIPLE conversational/clinical dimensions, not just synonym
      substitution
      ii. Vary: information type, verbal detail, clinical implication, medical indicator
      iii. Each option must lead to a DIFFERENT clinical interpretation or prognosis
   3. AGGRESSIVE WORD REPETITION WITH RECOMBINATION
      i. Use the SAME key structural words across MULTIPLE options but in DIFFERENT combinations
     ii. Each key descriptor should appear in 4-6 different options, paired with different qualifiers
      iii. The CORRECT answer is distinguished by its unique COMBINATION, not by unique words
      iv. This forces the model to understand the FULL description, not just match keywords
   4. OBSERVABLE PHENOMENA
      i. Ask about OBSERVABLE CONVERSATIONAL CONTENT (statements, exchanges, information shared),
      not subjective qualities
      ii. Options should describe what can be HEARD in the dialogue, not vague impressions
      iii. Focus on concrete conversational details rather than abstract characterizations
   5. CLINICAL GROUNDING
      i. All options must be grounded in medical reality
      ii. Each option MUST describe some part of the conversation
      iii. Options MUST be directly related to the dialogue content
   --------------------------------------------------
   D. MULTIPLE-CHOICE RULES
   --------------------------------------------------
   1. Exactly ten options: (a), (b), (c), (d), (e), (f), (g), (h), (i), (j)
   2. Exactly one correct answer
   3. Correct answer position MUST be randomly distributed across (a)-(j) - avoid clustering
   4. OPTION ORDERING RULE:
      i. Options with similar conversational descriptors must NOT be adjacent
      ii. Shuffle option order so that similar options are scattered
   5. Incorrect options may reflect:
      i. Patient's verbal claim taken at face value
      ii. Common medical assumption that conversation disproves
      iii. Correct conversational observation but wrong clinical interpretation
      iv. Wrong conversational detail but plausible clinical interpretation
   6. ANSWER OPTIONS DESIGN PRINCIPLE (STRICT)
      i. Use your thought process to come up with close, contrastive options that are similar to the
      correct option but are still incorrect
      ii. Repeat words AGGRESSIVELY across options to obfuscate. Options MUST be confusing
      iii. AVOID overt conversational cues - focus on subtle verbal details
      iv. DO NOT include temporal markers, durations, or time-based descriptors
   --------------------------------------------------
   E. QUESTION DESIGN PRINCIPLES
   --------------------------------------------------
   1. Questions under 25 words - direct, no clinical front-loading
   2. Ask about what the conversational evidence reveals, not just what was explicitly said
   3. Frame questions around contradiction, revelation, or hidden information
   4. DO NOT include temporal references, time markers, or duration-based questions
   THE CORRECT ANSWER must:
   - Identify the ACTUAL conversational detail present in the dialogue
   - Correctly interpret its clinical significance
   - Contradict or complicate the surface-level narrative when appropriate
   WRONG ANSWERS must:
   - Miss the conversational cue (trust surface claims instead)
   - Mischaracterize the verbal content
   - Misinterpret clinically
   - Mistake normal statements for pathology OR pathology for normal statements
   --------------------------------------------------
   F. QUALITY VERIFICATION
   --------------------------------------------------
   Before finalizing, verify each question passes:
   1. TEXT-ONLY TEST: Transcript alone cannot eliminate any option
   2. TEXTBOOK TEST: Medical knowledge alone cannot identify correct answer
   3. OPTION-LENGTH TEST: Each option is 6-12 words with proper structure
   4. MULTI-DIMENSION TEST: Options differ in multiple aspects, not just synonyms
   5. CLINICAL-DIFFERENCE TEST: Each option leads to different clinical interpretation
   --------------------------------------------------
```

```
    G. FAILURE CONDITION
    -------------------------------------------------
    If the conversation does NOT contain meaningful verbal evidence that could contradict or inform
    beyond the surface content, return None.
    -------------------------------------------------
    H. NAME AND IDENTIFIER CONSTRAINT (DE-IDENTIFICATION)
    -------------------------------------------------
    1. De-identification Requirement
      i. The question text, answer, and all options must be fully de-identified
      ii. Do NOT include any personal names, honorifics, initials, or unique identifiers
      iii. Do NOT reference specific clinicians, patients, locations, institutions, or dates
      iv. Use only generic role references such as "the patient" and "the clinician"
    -------------------------------------------------
    I. INPUT
    -------------------------------------------------
    The medical conversation containing speech dialogue: <AUDIO_INPUT>
    -------------------------------------------------
    J. OUTPUT FORMAT (STRICTLY RETURN a single parseable JSON array. No text outside JSON.)
    -------------------------------------------------
    [
      {
        "difficulty": "easy",
        "question": "...",
        "answer": "The answer is (X).",
        "options": "(a) ...\n(b) ...\n(c) ...\n(d) ...\n(e) ...\n(f) ...\n(g) ...\n(h) ...\n(i)
        ...\n(j) ..."
      },
      {
        "difficulty": "medium",
        "question": "...",
        "answer": "The answer is (X).",
        "options": "(a) ...\n(b) ...\n(c) ...\n(d) ...\n(e) ...\n(f) ...\n(g) ...\n(h) ...\n(i)
        ...\n(j) ..."
      },
      {
        "difficulty": "hard",
        "question": "...",
        "answer": "The answer is (X).",
        "options": "(a) ...\n(b) ...\n(c) ...\n(d) ...\n(e) ...\n(f) ...\n(g) ...\n(h) ...\n(i)
        ...\n(j) ..."
      }
    ]
```

### A.11.6. PROMPT FOR SPEECH PLUS SOUND QA PAIR GENERATION

```
medical_sound_speech_qa_generation:
  description: "Generate adversarial QA pairs requiring genuine acoustic reasoning from the given
  medical audio conversations."
  prompt: |
    You are an AI assistant specialized in medical audio understanding.

    You will be given a medical audio recording containing a conversation between a patient and a
    clinician, with embedded acoustic sounds.

    The audio is the ONLY source of truth.

    -------------------------------------------------
    A. OBJECTIVE
    -------------------------------------------------
    Generate THREE multiple-choice medical reasoning questions (MCQs) that require understanding:
    1. The conversation context (what was discussed, claimed, or described)
    2. The acoustic evidence (vocal quality, breathing patterns, coughs, pauses, sighs, etc.)
```

3. The clinical interpretation (what the acoustic evidence reveals about the patient's true state)

Each question must require the model to:
1. Listen to the actual audio (not just answering based on transcript)
2. Identify acoustic features that may contradict verbal claims
3. Integrate acoustic evidence with clinical reasoning to infer the correct answer
4. The questions should be waveform agnostic. No temporal or frequency characteristic should be given
5. The questions MUST be relevant to a medical practitioner for prognosis

The questions must increase in difficulty:
1. EASY: Requires subtle acoustic feature identification with all options appearing similar
2. MEDIUM: Requires acoustic feature identification and correlation of acoustic evidence across conversation with all options appearing nearly identical
3. HARD: Requires acoustic feature identification, correlation of acoustic evidence across conversation and extremely subtle acoustic distinctions with all options appearing nearly identical

```
--------------------------------------------------
B. CRITICAL CONSTRAINTS
--------------------------------------------------
```

1. ANTI-HALLUCINATION PRINCIPLE
   i. If the patient verbally claims distress, anxiety, or severity, the CORRECT answer must
   describe the ACTUAL acoustic quality heard
   ii. If the patient claims to be "fine" or "stable," look for hidden acoustic distress markers
   iii. The model MUST base answers on what is HEARD, not what is SAID

2. ACOUSTIC CORRELATION
   i. MEDIUM and HARD questions MAY rely on information from different parts of the conversation
   ii. For HARD questions, require correlation of acoustic evidence with dialogue content
   iii. Do NOT restate evidence in the question - require careful listening

```
--------------------------------------------------
C. OPTION FORMAT REQUIREMENTS (STRICT)
--------------------------------------------------
```

1. OPTION LENGTH AND STRUCTURE
   i. Each option MUST be 6-12 words minimum
   ii. Each option MUST follow structure: [acoustic observation] + [clinical qualifier/context]
   iii. DO NOT use single-word or two-word descriptors - expand to full descriptions

2. MULTI-DIMENSIONAL DIFFERENTIATION
   i. Options must differ in MULTIPLE acoustic/clinical dimensions, not just synonym substitution
   ii. Vary: sound type, acoustic quality, clinical implication, physiological indicator
   iii. Each option must lead to a DIFFERENT clinical interpretation or prognosis

3. AGGRESSIVE WORD REPETITION WITH RECOMBINATION
   i. Use the SAME key structural words across MULTIPLE options but in DIFFERENT combinations
   ii. Each key descriptor should appear in 4-6 different options, paired with different qualifiers
   iii. The CORRECT answer is distinguished by its unique COMBINATION, not by unique words
   iv. This forces the model to understand the FULL description, not just match keywords

4. OBSERVABLE PHENOMENA
   i. Ask about OBSERVABLE ACOUSTIC PHENOMENA (sounds, events), not subjective qualities
   ii. Options should describe what can be HEARD, not vague impressions
   iii. Focus on concrete sound characteristics rather than abstract vocal qualities

5. CLINICAL GROUNDING
   i. All options must be grounded in medical reality
   ii. Each option MUST describe some part of the input audio
   iii. Options MUST be directly related to the conversation content

```
--------------------------------------------------
D. MULTIPLE-CHOICE RULES
```

```
--------------------------------------------------
1. Exactly ten options: (a), (b), (c), (d), (e), (f), (g), (h), (i), (j)
2. Exactly one correct answer
3. Correct answer position MUST be randomly distributed across (a)-(j) - avoid clustering

4. OPTION ORDERING RULE:
    i. Options with similar acoustic descriptors must NOT be adjacent
    ii. Shuffle option order so that similar options are scattered

5. Incorrect options may reflect:
    i. Patient's verbal claim taken at face value
    ii. Common medical assumption that audio disproves
    iii. Correct acoustic observation but wrong clinical interpretation
    iv. Wrong acoustic descriptor but plausible clinical interpretation

6. ANSWER OPTIONS DESIGN PRINCIPLE (STRICT)
    i. Use your thought process to come up with close, contrastive options that are similar to the
    correct option but are still incorrect
    ii. Repeat words AGGRESSIVELY across options to obfuscate. Options MUST be confusing
    iii. AVOID overt acoustic cues - focus on subtle acoustic features
    iv. DO NOT include temporal markers, durations, or time-based descriptors

--------------------------------------------------
E. QUESTION DESIGN PRINCIPLES
--------------------------------------------------

1. Questions under 25 words - direct, no clinical front-loading
2. Ask about what the acoustic evidence reveals, not what was said
3. Frame questions around contradiction, revelation, or hidden information
4. DO NOT include temporal references, time markers, or duration-based questions

THE CORRECT ANSWER must:
- Identify the ACTUAL acoustic feature present in the audio
- Correctly interpret its clinical significance
- Contradict or complicate the verbal/textual narrative when appropriate

WRONG ANSWERS must:
- Miss the acoustic cue (trust text instead)
- Mischaracterize the acoustic quality
- Misinterpret clinically
- Mistake normal speech for pathology OR pathology for normal speech

--------------------------------------------------
F. QUALITY VERIFICATION
--------------------------------------------------

Before finalizing, verify each question passes:
1. TEXT-ONLY TEST: Transcript alone cannot eliminate any option
2. TEXTBOOK TEST: Medical knowledge alone cannot identify correct answer
3. OPTION-LENGTH TEST: Each option is 6-12 words with proper structure
4. MULTI-DIMENSION TEST: Options differ in multiple aspects, not just synonyms
5. CLINICAL-DIFFERENCE TEST: Each option leads to different clinical interpretation

--------------------------------------------------
G. FAILURE CONDITION
--------------------------------------------------
If the audio does NOT contain meaningful acoustic evidence that could contradict or inform beyond
the verbal content, return None.

--------------------------------------------------
H. NAME AND IDENTIFIER CONSTRAINT (DE-IDENTIFICATION)
--------------------------------------------------

1. De-identification Requirement
```

```
        i. The question text, answer, and all options must be fully de-identified
        ii. Do NOT include any personal names, honorifics, initials, or unique identifiers
        iii. Do NOT reference specific clinicians, patients, locations, institutions, or dates
        iv. Use only generic role references such as "the patient" and "the clinician"

   --------------------------------------------------
   I. INPUT
   --------------------------------------------------

   The medical audio containing speech and acoustic sounds: <AUDIO_INPUT>

   --------------------------------------------------
   J. OUTPUT FORMAT (STRICTLY RETURN a single parseable JSON array. No text outside JSON.)
   --------------------------------------------------

   [
     {
       "difficulty": "easy",
       "question": "...",
       "answer": "The answer is (X).",
       "options": "(a) ...\n(b) ...\n(c) ...\n(d) ...\n(e) ...\n(f) ...\n(g) ...\n(h) ...\n(i)
       ...\n(j) ..."
     },
     {
       "difficulty": "medium",
       "question": "...",
       "answer": "The answer is (X).",
       "options": "(a) ...\n(b) ...\n(c) ...\n(d) ...\n(e) ...\n(f) ...\n(g) ...\n(h) ...\n(i)
       ...\n(j) ..."
     },
     {
       "difficulty": "hard",
       "question": "...",
       "answer": "The answer is (X).",
       "options": "(a) ...\n(b) ...\n(c) ...\n(d) ...\n(e) ...\n(f) ...\n(g) ...\n(h) ...\n(i)
       ...\n(j) ..."
     }
   ]
```

### A.11.7. PROMPT FOR OPEN ENDED (SPEECH) QA PAIR GENERATION

```
open_ended_speech_qa_generation:
  description: "Generate adversarial open-ended QA pairs requiring genuine medical conversation
  reasoning from the given medical dialogues."
  template: |
    You are an AI assistant specialized in medical conversation understanding.
    You will be given a medical conversation recording between a patient and a clinician.
    Your **goal** is to create questions that FAIL models relying on shortcuts or generic medical
    knowledge.
    The conversation is the ONLY source of truth.
    --------------------------------------------------
    A. OBJECTIVE
    --------------------------------------------------
    Generate THREE open-ended medical reasoning questions that require understanding:
    1. The conversation content (what was discussed, claimed, or described)
    2. The implicit information (what's suggested but not directly stated, contradictions, hesitations)
    3. The clinical interpretation (what the conversation reveals about the patient's true state)

    Each question must require the model to:
    1. Listen to the actual conversation (not just answering based on transcript)
    2. Identify SUBTLE implicit information or contradictions in verbal delivery
    3. Integrate conversational nuance with clinical reasoning
```

4. Distinguish between surface claims and underlying conversational evidence
5. The questions MUST be relevant to a medical practitioner for prognosis

The questions must increase in difficulty:
1. EASY: Requires identifying subtle verbal cues and their clinical implications
2. MEDIUM: Requires synthesizing multiple conversational aspects with clinical reasoning
3. HARD: Requires complex multi-dimensional synthesis of conversational evidence
-----------------------------------------------
B. CRITICAL CONSTRAINTS
-----------------------------------------------

1. ANTI-HALLUCINATION PRINCIPLE
   i. If patient verbally claims distress/severity, the CORRECT answer must describe ACTUAL
   conversational evidence (hesitation, word choice, delivery patterns)
   ii. If patient claims to be "fine"/"stable," answer must identify hidden verbal distress markers
   iii. Answer MUST distinguish surface claims from underlying verbal evidence

2. CONVERSATIONAL NUANCE REQUIREMENT
   i. Questions should appear answerable with medical knowledge but REQUIRE specific conversational
   evidence
   ii. MEDIUM and HARD questions MUST synthesize information from different conversation parts
   iii. Ground truth must focus on SUBTLE verbal cues that generic models will miss

3. GROUNDING REQUIREMENT (ABSOLUTE)
   i. EVERY element in the question MUST reference something ACTUALLY PRESENT in the audio
   ii. EVERY element in the ground truth MUST be traceable to SPECIFIC conversational moments
   iii. Do NOT infer clinical content that requires knowledge beyond what is HEARD
   iv. Questions must be answerable BY LISTENING to the audio, not by medical knowledge alone

4. MULTI-DIMENSIONAL SYNTHESIS
   i. Answers must integrate BOTH conversational evidence AND clinical interpretation
   ii. Answers must distinguish between what was explicitly stated vs implicitly revealed
   iii. Answers should identify contradictions, hesitations, or patterns models typically miss
-----------------------------------------------
C. QUESTION DESIGN PRINCIPLES (STRICT)
-----------------------------------------------
1. APPARENT BREADTH WITH HIDDEN SPECIFICITY
   i. Questions should APPEAR to invite comprehensive medical answers
   ii. But ground truth requires identifying SPECIFIC subtle conversational cues
   iii. Frame questions broadly enough that models will add generic content
   iv. Examples: "What does X suggest about severity and management needs?", "What clinical risk
   arises from..."

2. MULTI-PART COMPLEXITY
   i. Combine 2-3 dimensions in questions: severity + implications, evidence + interpretation,
   claim + contradiction
   ii. Use "and" to connect multiple aspects: "What do verbal cues and described restrictions
   suggest..."
   iii. Questions should require synthesis, not single-point answers

3. SUBTLE VERBAL CUE DEPENDENCY
   i. Ground truth must depend on catching subtle cues: hesitation patterns, word choice, absolute
   claims ("every one"), qualifiers ("used to be")
   ii. Questions should NOT explicitly mention these cues - require models to identify them
   iii. Correct answers require noticing delivery patterns (fillers, pauses, emphasis) that
   transcripts miss

4. INTERPRETIVE FRAMING
   i. Ask "What does X suggest/reveal/indicate about Y" where Y is a broad clinical concept
   ii. Use verbs: suggest, reveal, indicate, reflect, arise from, emerge from
   iii. Focus on implications, risks, concerns, perspectives, patterns

5. AVOID OVERLY NARROW QUESTIONS
   i. Do NOT anchor to single specific phrases only
   ii. Do NOT ask simple factual recall questions

    iii. Questions should require interpretation and synthesis, not just identification

6. LENGTH CONSTRAINTS
   i. Questions must not exceed 40 words
   ii. Answers must not exceed 100 words
-----------------------------------------------
D. ANSWER FORMAT REQUIREMENTS (STRICT)
-----------------------------------------------
1. ANSWER FLEXIBILITY
   i. Answers can range from concise (5-10 words) to detailed (up to 100 words maximum)
   ii. Length should match the complexity needed to capture the conversational evidence
   iii. Prioritize precision over length - say exactly what's needed, no more

2. ANSWER CONTENT FOCUS
   i. State the clinical interpretation based on conversational evidence
   ii. Identify SPECIFIC subtle verbal cues (hesitation, word choice, absolute claims, qualifiers) and what was said
   iii. Explain the clinical significance of these specific verbal patterns
   iv. Note contradictions between claims and evidence, or connect to broader clinical reality (when relevant)

3. SUBTLE EVIDENCE FOCUS
   i. Answers must highlight verbal delivery patterns: hesitation ("um", "yeah"), absolute claims ("every one"), temporal qualifiers ("used to be")
   ii. Answers must explain how DELIVERY contradicts or complicates surface content
   iii. Answers must identify what generic models miss: subtle word choice, implications in phrasing
   iv. Do NOT just cite what was said - explain HOW it was said and WHY that matters

4. CLINICAL NUANCE REQUIREMENT
   i. Answers must provide SPECIFIC clinical interpretations, not generic management lists
   ii. Focus on: urgency levels, risk profiles, clinical thresholds, generational shifts, predictability
   iii. Answers should identify precise clinical concerns: "hypovolemia", "inflammatory markers", "narrow therapeutic window", "subclinical markers"
   iv. AVOID generic additions: "assess hydration, rule out infections, monitor vitals" UNLESS specifically grounded in conversational evidence

5. CONTRAST WITH SURFACE CLAIMS
   i. When patient makes explicit claims, answers must identify evidence that contradicts or complicates
   ii. Distinguish between "patient claims X" vs "conversational evidence reveals Y"
   iii. Highlight gaps between presentation and underlying reality

6. DE-IDENTIFICATION
   i. Use only "the patient" and "the clinician" - no names, locations, institutions, dates
-----------------------------------------------
E. TRAP DESIGN FOR GENERIC MODELS
-----------------------------------------------
Design questions to trap models that:
1. Add generic medical knowledge not grounded in conversation
2. Miss subtle verbal delivery cues (hesitation, qualifiers, absolute claims)
3. Take surface claims at face value without examining contradictions
4. Provide comprehensive management lists instead of focused interpretation
5. Underestimate or overestimate severity without conversational evidence
6. Give "reasonable but wrong" answers that sound plausible
-----------------------------------------------
F. QUALITY VERIFICATION
-----------------------------------------------
Before finalizing, verify each QA pair passes:
1. TRAP TEST: Would a generic model add content not in ground truth? (GOOD if yes)
2. SUBTLETY TEST: Does ground truth depend on subtle verbal cues models typically miss? (REQUIRED)
3. BREADTH TEST: Does question appear to invite broader answers than ground truth requires? (GOOD if yes)
4. NUANCE TEST: Does answer distinguish surface claims from conversational evidence? (REQUIRED)

```
   5. SPECIFICITY TEST: Does ground truth provide precise clinical terms, not generic ones? (REQUIRED)
   6. CONTRADICTION TEST: Does answer identify contradictions or complications models miss? (GOOD if
   yes)
   -------------------------------------------------
   G. DIFFICULTY CALIBRATION
   -------------------------------------------------
   EASY: Combine verbal cues with clinical implications in a way that invites generic medical additions
   MEDIUM: Synthesize 2-3 conversational aspects where models will miss subtle connections
   HARD: Complex multi-dimensional questions where models give comprehensive but wrong answers

   H. INPUT
   -------------------------------------------------
   The medical conversation containing speech dialogue: <AUDIO_INPUT>
   -------------------------------------------------
   I. OUTPUT FORMAT (STRICTLY RETURN a single parseable JSON array. No text outside JSON.)
   -------------------------------------------------
   [
     {
       "difficulty": "easy",
       "question": "...",
       "answer": "..."
     },
     {
       "difficulty": "medium",
       "question": "...",
       "answer": "..."
     },
     {
       "difficulty": "hard",
       "question": "...",
       "answer": "..."
     }
   ]
```

A.11.8. PROMPT FOR OPEN ENDED (SPEECH PLUS SOUND) QA PAIR GENERATION

```
  open_ended_speech_sound_qa_generation:
    description: "Generate adversarial open-ended QA pairs requiring genuine medical conversation
    reasoning from medical dialogues with acoustic context (speech + respiratory/background sounds)."
    template: |
      You are an AI assistant specialized in medical conversation and acoustic understanding.
      You will be given a medical conversation recording between a patient and a clinician that includes
      both speech dialogue and acoustic characteristics (breath sounds, background medical sounds, etc.).
      Your **goal** is to create adversarial open-ended QA pairs that FAIL models relying on shortcuts
      or generic medical knowledge.
      The conversation and its acoustic context of the audio are the ONLY sources of truth.
      -------------------------------------------------
      A. OBJECTIVE
      -------------------------------------------------
      Generate THREE open-ended medical reasoning questions in increasing difficulty(easy,medium, hard)
      that require understanding:
      1. The conversation content
      2. The acoustic characteristics
      3. The implicit information
      4. The clinical interpretation

      Each question must require the model to:
      1. Listen to the actual conversation AND acoustic characteristics (not just answering based on
      transcript)
      2. Identify SUBTLE implicit information or contradictions in verbal delivery AND acoustic patterns
      3. Integrate conversational nuance, acoustic evidence, and clinical reasoning
      4. Distinguish between surface claims and underlying conversational/acoustic evidence
```

5. The questions MUST be relevant to a medical practitioner for prognosis

The questions must increase in difficulty:
1. EASY: Requires identifying subtle verbal/acoustic cues and their clinical implications
2. MEDIUM: Requires synthesizing multiple conversational and acoustic aspects with clinical reasoning
3. HARD: Requires complex multi-dimensional synthesis of conversational and acoustic evidence
-------------------------------------------------
B. CRITICAL CONSTRAINTS
-------------------------------------------------
1. ANTI-HALLUCINATION PRINCIPLE
  i. If patient verbally claims distress/severity, the CORRECT answer must describe ACTUAL conversational and acoustic evidence (hesitation, word choice, delivery patterns, breath sounds, respiratory patterns, etc)
  ii. If patient claims to be "fine"/"stable," answer must identify hidden verbal distress markers OR contradicting acoustic evidence
  iii. Answer MUST distinguish surface claims from underlying verbal and acoustic evidence
  iv. Acoustic evidence (breath patterns, respiratory patterns, background sounds, etc) must be ACTUALLY PRESENT in the audio, not inferred

2. CONVERSATIONAL AND ACOUSTIC NUANCE REQUIREMENT
  i. Questions should appear answerable with medical knowledge but REQUIRE specific conversational and acoustic evidence
  ii. MEDIUM and HARD questions MUST synthesize information from different conversation parts AND/OR multiple acoustic features
  iii. Ground truth must focus on SUBTLE verbal and acoustic cues that generic models will miss
  iv. Questions should leverage the interaction between what is SAID and what is HEARD acoustically

3. GROUNDING REQUIREMENT (ABSOLUTE)
  i. EVERY element in the question MUST reference something ACTUALLY PRESENT in the audio (speech OR acoustic characteristics)
  ii. EVERY element in the ground truth MUST be traceable to SPECIFIC conversational moments OR acoustic features
  iii. Do NOT infer clinical content that requires knowledge beyond what is HEARD (speech + sounds)
  iv. Questions must be answerable BY LISTENING to the audio (speech + acoustic characteristics), not by medical knowledge alone
  v. Acoustic features (breath sounds, background sounds, respiratory sounds, etc) must be VERIFIABLE in the audio, not assumed

4. MULTI-DIMENSIONAL SYNTHESIS
  i. Answers must integrate conversational evidence, acoustic evidence, AND clinical interpretation, etc.
  ii. Answers must distinguish between what was explicitly stated vs implicitly revealed through speech delivery vs revealed through acoustic characteristics
  iii. Answers should identify contradictions between verbal claims and acoustic evidence, or patterns models typically miss
-------------------------------------------------
C. QUESTION DESIGN PRINCIPLES (STRICT)
-------------------------------------------------
1. APPARENT BREADTH WITH HIDDEN SPECIFICITY
  i. Questions should APPEAR to invite comprehensive medical answers
  ii. But ground truth requires identifying SPECIFIC subtle conversational and acoustic cues
  iii. Frame questions broadly enough that models will add generic content

2. MULTI-PART COMPLEXITY
  i. Combine 2-3 dimensions in questions: severity + implications, evidence + interpretation, claim + contradiction, verbal cues + acoustic patterns, etc
  ii. Use "and" to connect multiple aspects
  iii. Questions should require synthesis of speech AND acoustic evidence, not single-point answers

3. SUBTLE VERBAL AND ACOUSTIC CUE DEPENDENCY
  i. Ground truth must depend on catching subtle cues:
     - Verbal: hesitation patterns, word choice, absolute claims ("every one"), qualifiers ("used to be")

```
      - Acoustic: breath patterns (labored, shallow, irregular), respiratory sounds (wheezing,
        crackles), background sounds (oxygen equipment, monitoring devices), etc
    ii. Questions should NOT explicitly mention these cues - require models to identify them
    iii. Correct answers require noticing delivery patterns AND acoustic characteristics that
    transcripts miss
    iv. Leverage contradictions between what patient SAYS vs what acoustic evidence REVEALS

  4. AVOID OVERLY NARROW QUESTIONS
    i. Do NOT anchor to single specific phrases or single acoustic features only
    ii. Do NOT ask simple factual recall questions
    iii. Questions should require interpretation and synthesis of multiple evidence types, not just
    identification

  5. LENGTH CONSTRAINTS
    i. Questions must not exceed 25 words
    ii. Answers must not exceed 100 words
-------------------------------------------------
D. ANSWER FORMAT REQUIREMENTS (STRICT)
-------------------------------------------------
1. ANSWER FLEXIBILITY
  i. Answers can range from concise (5-10 words) to detailed (up to 100 words maximum)
  ii. Length should match the complexity needed to capture the conversational and acoustic evidence
  iii. Prioritize precision over length - say exactly what's needed, no more

2. ANSWER CONTENT FOCUS
  i. State the clinical interpretation based on conversational AND acoustic evidence
  ii. Identify SPECIFIC subtle verbal cues (hesitation, word choice, absolute claims, qualifiers,
  etc) AND acoustic characteristics (breath patterns, respiratory sounds, background sounds, etc)
  and what was said/heard
  iii. Explain the clinical significance of these specific verbal and acoustic patterns
  iv. Note contradictions between verbal claims and acoustic evidence, or connect to broader
  clinical reality (when relevant)

3. SUBTLE EVIDENCE FOCUS
  i. Answers must highlight verbal delivery patterns: hesitation ("um", "yeah"), absolute claims
  ("every one"), temporal qualifiers ("used to be"), etc.
  ii. Answers must highlight acoustic patterns: breath characteristics (labored, shallow, rapid),
  respiratory sounds (wheezing, crackles, stridor), background sounds (oxygen delivery, medical
  equipment), etc
  iii. Answers must explain how DELIVERY and ACOUSTIC EVIDENCE contradicts or complicates surface
  content
  iv. Answers must identify what generic models miss: subtle word choice, implications in phrasing,
  acoustic contradictions
  v. Do NOT just cite what was said - explain HOW it was said, WHAT was heard acoustically, and WHY
  that matters

4. CLINICAL NUANCE REQUIREMENT
  i. Answers must provide SPECIFIC clinical interpretations, not generic management lists
  ii. Focus on: urgency levels, risk profiles, clinical thresholds, respiratory status,
  oxygenation concerns, etc
  iii. Answers should identify precise clinical concerns from acoustic evidence
  iv. AVOID generic additions UNLESS specifically grounded in conversational or acoustic evidence

5. CONTRAST WITH SURFACE CLAIMS
  i. When patient makes explicit claims, answers must identify verbal or acoustic evidence that
  contradicts or complicates
  ii. Distinguish between "patient claims X" vs "conversational evidence reveals Y" vs "acoustic
  characteristics indicate Z"
  iii. Highlight gaps between verbal presentation and underlying acoustic reality

6. DE-IDENTIFICATION
  i. Use only "the patient" and "the clinician" - no names, locations, institutions, dates
-------------------------------------------------
E. QUALITY VERIFICATION
```

```
--------------------------------------------------
Before finalizing, verify each QA pair passes:
1. SUBTLETY TEST: Does ground truth depend on subtle verbal and/or acoustic cues models typically
miss? (REQUIRED)
2. BREADTH TEST: Does question appear to invite broader answers than ground truth requires? (GOOD
if yes)
3. NUANCE TEST: Does answer distinguish surface claims from conversational and acoustic evidence?
(REQUIRED)
4. SPECIFICITY TEST: Does ground truth provide precise clinical terms, not generic ones? (REQUIRED)
5. CONTRADICTION TEST: Does answer identify contradictions or complications models miss (verbal vs
acoustic)? (GOOD if yes)
6. ACOUSTIC TEST: Does answer leverage acoustic evidence that transcripts cannot capture?
(REQUIRED for at least 2 out of 3 questions)
--------------------------------------------------
F. DIFFICULTY CALIBRATION
--------------------------------------------------
EASY: Combine verbal/acoustic cues with clinical implications in a way that invites generic
medical additions
MEDIUM: Synthesize 2-3 conversational and acoustic aspects where models will miss subtle
connections
HARD: Complex multi-dimensional questions integrating verbal claims, acoustic contradictions, and
clinical synthesis where models give comprehensive but wrong answers

G. INPUT
--------------------------------------------------
The medical conversation containing speech dialogue and acoustic characteristics (breath sounds,
background medical sounds): <AUDIO_INPUT>
--------------------------------------------------
H. OUTPUT FORMAT (STRICTLY RETURN a single parseable JSON array. No text outside JSON.)
--------------------------------------------------
[
  {
    "difficulty": "easy",
    "question": "...",
    "answer": "..."
  },
  {
    "difficulty": "medium",
    "question": "...",
    "answer": "..."
  },
  {
    "difficulty": "hard",
    "question": "...",
    "answer": "..."
  }
]
```

### A.11.9. PROMPT FOR SPEECH ONLY QA PAIR GENERATION

```
mcqs_speech_qa_generation:
    description: "Generate adversarial QA pairs requiring genuine medical conversation reasoning from
    the given medical dialogues."
    template: |
      You are an AI assistant specialized in medical conversation understanding.
      You will be given a medical conversation recording between a patient and a clinician.
      Your **goal** is to create questions that FAIL models relying on shortcuts.
      The conversation is the ONLY source of truth.
      --------------------------------------------------
      A. OBJECTIVE
      --------------------------------------------------
      Generate THREE multiple-choice medical reasoning questions (MCQs) that require understanding:
```

1. The conversation content (what was discussed, claimed, or described, etc)
2. The implicit information (what's suggested but not directly stated, contradictions, hesitations, etc)
3. The clinical interpretation (what the conversation reveals about the patient's true state)
Each question must require the model to:
1. Listen to the actual conversation (not just answering based on transcript)
2. Identify implicit information or contradictions in verbal statements
3. Integrate conversational evidence with clinical reasoning to infer the correct answer
4. The questions should focus on conversational content and clinical reasoning
5. The questions MUST be relevant to a medical practitioner for prognosis
The questions must increase in difficulty:
1. EASY: Requires subtle information identification from the conversation with all options appearing similar
2. MEDIUM: Requires information identification and correlation of evidence across conversation with all options appearing nearly identical
3. HARD: Requires information identification, correlation of evidence across conversation and extremely subtle conversational distinctions with all options appearing nearly identical
--------------------------------------------------
B. CRITICAL CONSTRAINTS
--------------------------------------------------
1. ANTI-HALLUCINATION PRINCIPLE
  i. If the patient verbally claims distress, anxiety, or severity, etc the CORRECT answer must describe the ACTUAL conversational evidence heard
  ii. If the patient claims to be "fine" or "stable," look for hidden verbal distress markers or contradictions
  iii. The model MUST base answers on what is HEARD in the conversation, not assumptions
2. CONVERSATIONAL CORRELATION
  i. MEDIUM and HARD questions MAY rely on information from different parts of the conversation
  ii. For HARD questions, require correlation of verbal evidence with dialogue content
  iii. Do NOT restate evidence in the question - require careful listening
--------------------------------------------------
C. OPTION FORMAT REQUIREMENTS (STRICT)
--------------------------------------------------
1. OPTION LENGTH AND STRUCTURE
  i. Each option MUST be 6-12 words minimum
  ii. Each option MUST follow structure: [conversational observation] + [clinical qualifier/context]
  iii. DO NOT use single-word or two-word descriptors - expand to full descriptions
2. MULTI-DIMENSIONAL DIFFERENTIATION
  i. Options must differ in MULTIPLE conversational/clinical dimensions, not just synonym substitution
  ii. Vary: information type, verbal detail, clinical implication, medical indicator
  iii. Each option must lead to a DIFFERENT clinical interpretation or prognosis
3. AGGRESSIVE WORD REPETITION WITH RECOMBINATION
  i. Use the SAME key structural words across MULTIPLE options but in DIFFERENT combinations
 ii. Each key descriptor should appear in 4-6 different options, paired with different qualifiers
  iii. The CORRECT answer is distinguished by its unique COMBINATION, not by unique words
  iv. This forces the model to understand the FULL description, not just match keywords
4. OBSERVABLE PHENOMENA
  i. Ask about OBSERVABLE CONVERSATIONAL CONTENT (statements, exchanges, information shared), not subjective qualities
  ii. Options should describe what can be HEARD in the dialogue, not vague impressions
  iii. Focus on concrete conversational details rather than abstract characterizations
5. CLINICAL GROUNDING
  i. All options must be grounded in medical reality
  ii. Each option MUST describe some part of the conversation
  iii. Options MUST be directly related to the dialogue content
--------------------------------------------------
D. MULTIPLE-CHOICE RULES
--------------------------------------------------
1. Exactly ten options: (a), (b), (c), (d), (e), (f), (g), (h), (i), (j)
2. Exactly one correct answer
3. Correct answer position MUST be randomly distributed across (a)-(j) - avoid clustering
4. OPTION ORDERING RULE:

```
      i. Options with similar conversational descriptors must NOT be adjacent
      ii. Shuffle option order so that similar options are scattered
   5. Incorrect options may reflect:
      i. Patient's verbal claim taken at face value
      ii. Common medical assumption that conversation disproves
      iii. Correct conversational observation but wrong clinical interpretation
      iv. Wrong conversational detail but plausible clinical interpretation
   6. ANSWER OPTIONS DESIGN PRINCIPLE (STRICT)
      i. Use your thought process to come up with close, contrastive options that are similar to the
      correct option but are still incorrect
      ii. Repeat words AGGRESSIVELY across options to obfuscate. Options MUST be confusing
      iii. AVOID overt conversational cues - focus on subtle verbal details
      iv. DO NOT include temporal markers, durations, or time-based descriptors
   --------------------------------------------------
   E. QUESTION DESIGN PRINCIPLES
   --------------------------------------------------
   1. Questions under 25 words - direct, no clinical front-loading
   2. Ask about what the conversational evidence reveals, not just what was explicitly said
   3. Frame questions around contradiction, revelation, or hidden information
   4. DO NOT include temporal references, time markers, or duration-based questions
   THE CORRECT ANSWER must:
   - Identify the ACTUAL conversational detail present in the dialogue
   - Correctly interpret its clinical significance
   - Contradict or complicate the surface-level narrative when appropriate
   WRONG ANSWERS must:
   - Miss the conversational cue (trust surface claims instead)
   - Mischaracterize the verbal content
   - Misinterpret clinically
   - Mistake normal statements for pathology OR pathology for normal statements
   --------------------------------------------------
   F. QUALITY VERIFICATION
   --------------------------------------------------
   Before finalizing, verify each question passes:
   1. TEXT-ONLY TEST: Transcript alone cannot eliminate any option
   2. TEXTBOOK TEST: Medical knowledge alone cannot identify correct answer
   3. OPTION-LENGTH TEST: Each option is 6-12 words with proper structure
   4. MULTI-DIMENSION TEST: Options differ in multiple aspects, not just synonyms
   5. CLINICAL-DIFFERENCE TEST: Each option leads to different clinical interpretation
   --------------------------------------------------
   G. FAILURE CONDITION
   --------------------------------------------------
   If the conversation does NOT contain meaningful verbal evidence that could contradict or inform
   beyond the surface content, return None.
   --------------------------------------------------
   H. NAME AND IDENTIFIER CONSTRAINT (DE-IDENTIFICATION)
   --------------------------------------------------
   1. De-identification Requirement
      i. The question text, answer, and all options must be fully de-identified
      ii. Do NOT include any personal names, honorifics, initials, or unique identifiers
      iii. Do NOT reference specific clinicians, patients, locations, institutions, or dates
      iv. Use only generic role references such as "the patient" and "the clinician"
   --------------------------------------------------
   I. INPUT
   --------------------------------------------------
   The medical conversation containing speech dialogue: <AUDIO_INPUT>
   --------------------------------------------------
   J. OUTPUT FORMAT (STRICTLY RETURN a single parseable JSON array. No text outside JSON.)
   --------------------------------------------------
   [
     {
       "difficulty": "easy",
       "question": "...",
       "answer": "The answer is (X).",
```

```
        "options": "(a) ...\n(b) ...\n(c) ...\n(d) ...\n(e) ...\n(f) ...\n(g) ...\n(h) ...\n(i)
        ...\n(j) ..."
      },
      {
        "difficulty": "medium",
        "question": "...",
        "answer": "The answer is (X).",
        "options": "(a) ...\n(b) ...\n(c) ...\n(d) ...\n(e) ...\n(f) ...\n(g) ...\n(h) ...\n(i)
        ...\n(j) ..."
      },
      {
        "difficulty": "hard",
        "question": "...",
        "answer": "The answer is (X).",
        "options": "(a) ...\n(b) ...\n(c) ...\n(d) ...\n(e) ...\n(f) ...\n(g) ...\n(h) ...\n(i)
        ...\n(j) ..."
      }
    ]
```

## A.12. Reproducibility Details

To ensure reproducibility and fair cross-model comparison, we specify the inference constraints used throughout our benchmarking experiments.

All audio inputs were resampled to 16 kHz prior to inference across all 13 models. For synthetically generated audio (Speech+Sound and Voice QA categories), the native generation rate was 44.1 kHz via ElevenLabs v3, which was subsequently downsampled to 16 kHz to maintain consistency with the open-source audio sources. Short-form clinical conversations (Speech Only and Speech+Sound) were up to 3 minutes in length, Long-Form audio strictly exceeded 3 minutes, and sound-only waveforms (heart, lung, and cough sounds) were roughly 30 seconds in length. All durations fell within the context window limits of every evaluated model, and no audio chunking or segmentation was applied during inference. Identical inference-time prompt templates were used across all 13 models for each QA pair type. The QA generation prompts used to construct the benchmark are provided in Appendix A.11, and the acoustic tag insertion prompts used during synthetic audio generation are provided in Appendix A.7.

