# OpenReview forum: "MedMosaic: A Challenging Large Scale Benchmark of Diverse Medical Audio"
_ICML.cc/2026/Conference — ICML 2026 regular_

### Official Review · Reviewer_bz74 · 2026-03-10

**Soundness:** 2
**Presentation:** 3
**Significance:** 4
**Originality:** 3
**Overall Recommendation:** 5
**Confidence:** 4

**Summary:**

Medical health data is typically difficult and expensive to obtain due to necessary protections and privacy, increasing the difficulty to train LLMs in the field of medicine. The authors propose MedMosaic, a synthetic medical QA dataset which alleviates the costs and security required with real world health datasets. This dataset is split into multiple pieces, being sound-only (e.g. coughs and wheezes,) speech only (language without any medical events,) speech-sound pairs which combine the raw language with different medical events such as coughing, sniffling, or whistling, and multi-turn which require multiple steps of reasoning. These pieces together compose a large dataset which can train benchmark LLMs to pick up on lexical and paralinguistic cues, as well as develop its CoT process. The task of training LLMs to classify medical issues is still a difficult task, with its best model performing at just under 70% accuracy, but invites further research into the curation of open-source synthetic medical datasets.

**Compliance With Llm Reviewing Policy:**

Affirmed.

**Ethical Review Concerns:**

With the incorporation of the authors' comments, the strength of the manuscript has increased highly and I encourage acceptance. Medical classifier models are an important innovation within the field, and the incorporation of LLMs and other methods like RAG introduce more ways for medical knowledge to be applied both in academia and in practice. Here, the ability to classify different medical conditions solely based on audio - whether textual or paralinguistic - is a major advancement for both medicine and natural language processing. There still remain some comments, which mostly are regarding phrasing or overall narrative flow, however I do not feel these comments are as important as the ones addressed in the rebuttals, and can be refined with more time.

**Final Justification:**

With the incorporation of the authors' comments, the strength of the manuscript has increased highly and I encourage acceptance. Medical classifier models are an important innovation within the field, and the incorporation of LLMs and other methods like RAG introduce more ways for medical knowledge to be applied both in academia and in practice. Here, the ability to classify different medical conditions solely based on audio - whether textual or paralinguistic - is a major advancement for both medicine and natural language processing. There still remain some comments, which mostly are regarding phrasing or overall narrative flow, however I do not feel these comments are as important as the ones addressed in the rebuttals, and can be refined with more time.

**Key Questions For Authors:**

1. How do you ensure that the generated questions cannot be answered by the LLMs general knowledge?

2. Was distractor obfuscation relatively uniform across the QA dataset. I.e. were the questions  generated according to a distribution of some sort of similarity metric? Is it possible that some of these questions could be, by chance, generated to be correct?

3. Were you able to obtain results of these same benchmark models on real medical data, whether sound only or speech only, as a comparison to your dataset? The impact of this paper would be greatly increased if you are able to provide a comparison of at least some of the tested models on a real medical dataset (i.e. showing improvement or marginal comparable results.)

4. How was the difficulty of the questions determined? Labels of "easy," "medium," and "hard" are given but it is not explained how these difficulties were verifed, or arranged to be of the same scale (i.e. are all "easy" questions truly easy?)

**Limitations:**

yes

**Strengths And Weaknesses:**

Strengths
1. Synthetic datasets do not require the same level of protections and privacy of real world medical data (such as IRB,) and as such can be continuously expanded without extra consideration. This also saves  time (of both the researching parties and the participants) and money, allowing quick and instant access to tens of thousands of relevant medical data.

2. Tests on numerous different LLMs, although showing varying results, indicate that the dataset has potential to perform well depending on model architecture or other considerations. This large testing grid (of different question types like sound only or speech only, against thirteen different SASO LLMs) demonstrates a strong push towards the generalizability of the dataset on performance.

Weaknesses
1. No comparisons. The authors present results for their synthetic dataset across numerous benchmark LLMs, but do not evaluate against a real-life dataset. It is presented as a key limitationthat the purpose of this dataset is to bypass the difficulties of obtaining a large amount of medical health data, however no mention as to the performance compared to real medical health data is made. This should be a primary result to demonstrate the advantage of using such a synthetic dataset, that it yields comparable, or at most, slightly worse results than a real and expensive dataset.

2. Overall paper flow. Phrasing in numerous sections throughout the manuscript seem either out of order or redundant, disrupting understanding. For example, the abstract instantly jumps into the contribution of the manuscript, but then takes a step back to reference the challenge typical medical audio datasets, to then refer back to MedMosaic. I personally think it would be more clear to the reader if the problems and consequences were laid out first, and then present the solution that MedMosaic offers.

Another example is during Section 3.1 explaining the QA pairs. Specifically, when explaining how the questions are generated to only have one answer, and then following that phrase with "We generate three questions of varying difficulty levels." Also, giving the descriptions of Easy, Medium, and Hard implicitly suggest increased level of sophisticated questions, removing the need for that explicit mention (or pershaps, 'We generate three questions at increasing levels of difficulty.")

---

> ### Author Rebuttal · Authors · 2026-03-31
>
> ## W1
>
> We thank the reviewer for the suggestion and acknowledge that the performance in relation to real medical datasets is not clearly stated. This is due to the unique structure of MedMosaic.
>
> Our 2-tiered dataset has:
>
> - **5,592 real-life** and **11,223 synthetic** conversations generated from real transcripts, closely approximating realistic clinical scenarios (details in Appendix A.1).
> - **46,701 fully synthetic QA pairs** designed to probe reasoning. Our motivation for devising QA pairs synthetically is to achieve scaling beyond the capabilities of human curators.
>
> To the best of our knowledge, no existing manually curated QA dataset spans all QA varieties included in MedMosaic. CaReSound (Wang et al., 2025) includes heart and lung sounds with QA pairs but is limited to short recordings without clinical speech or multi-turn reasoning. RA-QA (Bertolino et al., 2026) focuses on respiratory audio from multiple sources but excludes cardiac audio and clinical conversations.
>
> None of these datasets combine clinical dialogue, physiological sounds, long recordings, and multi-turn reasoning, which forms the core modalities in MedMosaic and where we observe significant performance gaps. Thus, a direct comparison with a real-life QA dataset is not feasible.
>
> ---
>
> ## W2
>
> Great suggestion! We will restructure the paper as follows:
>
> - The **abstract** will be restructured to first motivate the problem — including privacy constraints, the underrepresentation of complex medical audio scenarios, and the gap in multi-turn evaluation — before introducing MedMosaic.
> - The **introduction** will further motivate MedMosaic, elaborate existing challenges, and provide an overview of our solution in parallel to the corresponding challenge it addresses.
> - In **Section 3.1**, the sentence regarding questions having a single correct answer will be moved to follow the description of difficulty levels, resolving the ambiguity noted by the reviewer. The difficulty labels will also be revised to read "at increasing levels of difficulty."
>
> ---
>
> ## Q1
>
> This is an insightful question. We address the concern as follows:
>
> - Our QA generation prompts explicitly instruct Gemini-3-Flash that any question answerable from medical knowledge alone is invalid (Appendix A.7). This encourages the model to treat the audio as the sole source of truth during QA construction. We inspect Gemini's CoT to ensure the prompt instruction is being followed at inference time.
> - SMEs rate each QA pair on an **"Audio Necessity"** parameter scale from 1 to 3. As reported in Table 10, **98.6%** of pairs received the highest rating, meaning experts confirmed that the questions require listening to the audio.
> - In view of suggestions from multiple reviewers, we will perform a **"no-audio" ablation** requiring models to answer questions without any input audio. We expect accuracies to drop considerably, demonstrating strict audio dependence.
>
> ---
>
> ## Q2
>
> Thanks for probing distractor design.
>
> - Distractor generation uses shared prompt templates applied consistently across the dataset.
> - Templates enforce three key properties for incorrect options:
>   - (a) Clinically plausible
>   - (b) Acoustically confusable with the correct answer
>   - (c) Distinguishable only through careful reasoning over audio
> - Prompts encourage lexical overlap between options to require fine-grained discrimination.
> - Only one correct answer is enforced per question to prevent ambiguity (Appendix A.7).
> - Validated by SME pipeline: **72.6%** of distractors received the highest **"Distractor Quality"** rating.
>
> ---
>
> ## Q3
>
> Please see our response to W1.
>
> ---
>
> ## Q4
>
> Difficulty is calibrated through the prompts in Appendix A.7. We have the following levels, generated by prompting Gemini-3-Flash successively:
>
> | Level  | Description |
> |--------|-------------|
> | **Easy**   | Answers require direct retrieval from input audio |
> | **Medium** | Requires temporal correlation over distant information |
> | **Hard**   | Requires inferring medical condition progression implicitly |
>
> Experts rated **83.9%** of questions as high quality on the **"Question Quality"** parameter, strengthening our difficulty claim. In the camera-ready version, we will report model accuracy stratified by difficulty level across QA categories.

---

> > ### Author Rebuttal · Reviewer_bz74 · 2026-04-03
> >
> > Thank you for your clear and informative response. Many of my concerns came from misunderstandings, for which I apologize. I feel much more confident in understanding your work now, and believe my concerns are largely addressed. As your datset does include real-world data, I am inclined to trust the performance more, although a direct comparison against some form of medical audio dataset (such as a patient transcript, or maybe even sounds of medical events like coughing from TV shows) would add even more reach. Further, the questions of distractors and audio necessity were adequately answered, and I can more readily find these answers within the manuscript. I will be updating my scores accordingly.

---

> > > ### Author Response · Authors · 2026-04-04
> > >
> > > We sincerely thank the reviewer for the thoughtful follow-up and for carefully reconsidering our responses.
> > >
> > > We also thank you for the constructive suggestion regarding evaluation on medical audio data. We agree that extending our analysis to include such domain-specific datasets (e.g., patient transcripts or clinically relevant audio events) would further strengthen the applicability and impact of the work, and we will include this as an important direction for future investigation.
> > >
> > > We are encouraged that the clarifications regarding distractors and the role of audio were helpful, and we will ensure that these aspects are made even clearer in the final version of the manuscript.
> > >
> > > Thank you again for your valuable feedback and for updating your assessment.

---

### Official Review · Reviewer_vtBc · 2026-03-13

**Soundness:** 3
**Presentation:** 4
**Significance:** 3
**Originality:** 3
**Overall Recommendation:** 4
**Confidence:** 4

**Summary:**

The authors introduce MedMosaic, a large-scale multimodal benchmark designed to evaluate the reasoning capabilities of large audio-language models on medical audio. The dataset contains 46,701 QA pairs covering diverse audio types, including isolated physiological sounds, clinical conversations and mixed audio containing both speech and clinical artifacts. To overcome the scarcity of real-world medical audio, the authors utilize a scalable synthetic generation pipeline powered by LLMs and TTS models.

**Compliance With Llm Reviewing Policy:**

Affirmed.

**Final Justification:**

The rebuttal clarified the points I had raised.

**Key Questions For Authors:**

1. The benchmark focuses on a predefined set of physiological sounds and utilizes a curated pool of synthetic voices. How can the TTS synthetic pipeline scale to cover a much longer tail of diverse, complex, or rare vocal patterns (such as the gradual acoustic degradation over time)?
2. Could you provide a performance breakdown comparing how the models performed on the purely synthetic subsets versus the real-world datasets included in your benchmark?

**Limitations:**

In real-world medicine, patient care is longitudinal. Conditions evolve over weeks, months, or years. Clinicians don't just look at a single snapshot; they rely on multi-step forward prediction to figure out a patient's risk trajectory based on how their symptoms and acoustic biomarkers progress over multiple visits. In the paper, they are only talking about tracking context within a single 3-minute recording or connecting clues across a short 3-turn dialogue sequence.

**Strengths And Weaknesses:**

Strengths
1. This paper evaluated 13 diverse models across different architectural families (LALMs, LARMs, multimodal foundation models) provides a highly valuable snapshot of current state-of-the-art capabilities and highlights severe modality biases.
2. The way the multi-turn questions are set up is novel, making sure the models actually have to maintain context and track the state across turns, rather than just pattern-matching.

Weaknesses
1. While the multi-turn setup is great for short-term memory, real clinical diagnosis is fundamentally about longitudinal progression.
2. The models might be overfitting to TTS generation artifacts instead of learning genuine clinical acoustics.

---

> ### Author Rebuttal · Authors · 2026-03-31
>
> ## W1
>
> While we acknowledge the reviewer's feedback that medical diagnosis is a longitudinal progression, we would also like to emphasize that reasoning intelligently over a single visit conversation is also a cumbersome problem. Thus, in this work we tackle the problem of testing reasoning capabilities — especially over temporally distant information relayed in long-form audio and multi-turn questions based on it.
>
> We searched for multiple clinical sessions from the same doctor-patient pair to model longitudinal progression, but were unable to find any such publicly available dataset. In future work, we will extend MedMosaic to cover longitudinal data, as we are building a real system and not solving an academic problem. We thank the reviewer for their pragmatic suggestion.
>
> ---
>
> ## W2
>
> We thank the reviewer for this insightful remark and acknowledge that synthetic artifacts may deviate slightly from real clinical acoustics. We had to resort to TTS models because speech data with medical artifacts like sneezes and coughs is scarce. However, we also include a substantial share of real-life physiological and paralinguistic sounds to achieve a well-rounded dataset of genuine clinical acoustics (details in Appendix A.1). This prevents our accuracy metrics from overfitting to synthetically generated clinical acoustics.
>
> ---
>
> ## Q1
>
> We thank the reviewer for this insightful question. We scale the TTS synthetic pipeline to cover a much longer tail of diverse, complex, or rare vocal patterns as follows:
>
> 1. ElevenLabs' flexibility allows access to a wide variety of voices with different accents, including multilingual speakers who can naturally switch between languages based on the transcript.
> 2. Complex traits such as emotional expression, changes in pacing and rhythm, and subtle vocal textures like breathiness are modelled by placing acoustic tags directly in the transcript.
> 3. Our dataset used **35 tags** (22 with direct medical relevance) as a reference set, but the pipeline is not constrained to this list — it is designed to place tags contextually using an LLM.
> 4. Our dynamic tag placement strategy using an LLM is verified by SMEs for its efficacy.
> 5. The full tag list and generation prompts will be included in the appendix for transparency and reproducibility.
>
> ---
>
> ## Q2
>
> We classify QA pair audio types into two categories based on the source of transcript (omitting Open-Ended):
>
> - **Real:** MCQ Speech, MCQ Sound, Multi-Turn, Long Form
> - **Synthetic:** MCQ Speech-Sound, Voice QA
>
> Gemini 2.5 Pro and Flash perform better or maintain stability on synthetic datasets, whereas other models (such as Qwen and Gemma) see a significant drop (up to 10%) on synthetic tasks. Thus, Gemini is resilient to audio tags injected during synthetic data generation. The following table summarizes our findings, and we will include this in the camera-ready version of the paper.
>
> | Model             | Real (%) | Synthetic (%) |
> |-------------------|----------|---------------|
> | Gemini 2.5 Pro    | 65.3     | 68.6          |
> | Gemini 2.5 Flash  | 57.7     | 59.1          |
> | Qwen Omni 7B      | 44.5     | 32.1          |
> | Gemma 3n 8B       | 42.7     | 30.0          |
> | Desta25 Audio     | 39.7     | 36.6          |
> | Baichuan Omni     | 37.0     | 32.6          |
> | Phi4 MM           | 38.0     | 27.2          |
> | Kimi Audio        | 37.0     | 25.2          |
> | GPT-4o Audio      | 29.3     | 36.4          |
> | Audio Reasoner    | 31.5     | 26.4          |
> | Audio Flamingo 3  | 23.2     | 10.8          |
> | GAMA              | 24.9     | 12.3          |
> | R1 AQA            | 20.5     | 12.9          |

---

> > ### Author Rebuttal · Reviewer_vtBc · 2026-04-03
> >
> > Thank you for your detailed response. The rebuttal clarified the points I had raised.

---

### Official Review · Reviewer_pNbf · 2026-03-14

**Soundness:** 3
**Presentation:** 3
**Significance:** 3
**Originality:** 3
**Overall Recommendation:** 4
**Confidence:** 4

**Summary:**

The paper introduces MedMosaic, a large-scale benchmark for medical audio question answering that spans physiological sounds (heart, lung, cough), clinical dialogues, mixed speech+nonverbal sounds, multi-turn and long-form contexts, open-ended generation, and voice-embedded (spoken) questions. The authors propose a synthetic generation pipeline that integrates controlled TTS speech and inserted physiological/acoustic artifacts, producing 46,701 QA pairs, and benchmark 13 contemporary audio/multimodal reasoning models, finding the task remains challenging (best weighted average ≈68.1% with Gemini‑2.5‑Pro). Limited expert review suggests most items are clinically acceptable, and the analysis highlights modality biases (speech vs sound) and question-type sensitivities across models.

**Compliance With Llm Reviewing Policy:**

Affirmed.

**Key Questions For Authors:**

1. The multi-regime design (sound-only, speech-only, speech+sound, multi-turn, long-form, voice QA, open-ended) is well-motivated and aligns with clinical reality where cues are distributed over time and modalities.
2. Reliance on LLM-generated gold answers (MCQ) and LLM-judged scoring (open-ended) risks circularity and stylistic bias; explicit calibration against human-graded subsets would improve validity.
3. The authors acknowledge leakage via question semantics and parsing fragility; introducing audio-masked controls and a “no-audio” baseline would quantify how often items are solvable without acoustic evidence.

**Limitations:**

yes

**Strengths And Weaknesses:**

Strengths:
1. The benchmark unifies multiple medical audio regimes—physiological auscultation, clinical dialogue, and mixed audio—within a single evaluation suite, including less-explored voice-embedded QA and multi-turn chains.
2. The synthetic generation pipeline offers controllable insertion of paralinguistic and physiological cues (e.g., coughs, wheeze, prosody), enabling targeted reasoning probes that are hard to source at scale with real patient data.

Weaknesses:
1. Ground-truth answers for MCQs are generated by an LLM (Gemini‑3‑Flash), introducing potential annotation bias and error propagation; for open-ended tasks, both reference answers and the judge are LLM-driven, compounding dependency.
2. The synthetic audio (TTS + inserted artifacts) risks domain shift from real clinical acoustics; only a small fraction is validated by SMEs, and validation did not include overlapping reviews to measure inter-rater reliability.
3. Inference constraints (audio length limits, chunking strategy, sample rate, prompt templates) are under-specified, making fair cross-model comparisons and reproducibility difficult.

---

> ### Author Rebuttal · Authors · 2026-03-31
>
> ## W1
>
> Great insight! To analyze bias and error we treat MCQs and open-ended questions separately.
>
> 1. For **MCQs**, we guide the QA generation process such that there is only one correct option, and undertake SME validation to guard against incorrect ground truth.
> 2. For **OE questions**, we select the judge LLM as **GPT-5.1**, which is from a different family than Gemini, meaning that it is largely impartial — mitigating compounding error and dependencies.
> 3. As a future direction, we will generate QA pairs from multiple LLMs and check whether there is a pattern to the bias. If Gemini continues to outperform other models in this scenario as well, it will establish its robustness to source LLM.
> 4. However, we agree that calibration against human-graded subsets would strengthen credibility, and we will work towards a more comprehensive and exhaustive human validation.
> 5. We will report **Fleiss' κ** between Gemini-3-Flash-generated MCQ answers and human-assigned correct options on a calibration subset, and **Spearman correlation** between GPT-5.1 judge scores and human grader scores for open-ended evaluation in the camera-ready version, keeping in line with your recommendation.
>
> ---
>
> ## W2
>
> We appreciate this observation. The 145-pair SME validation was stratified across all categories and difficulty levels, but we recognize it represents a small fraction of 46,701 pairs and lacks inter-rater reliability measurement.
>
> On domain shift: our pipeline uses **ElevenLabs v3** with **151 naturally sounding voices** curated for demographic diversity and clinical acoustic suitability. We acknowledge that a gap between TTS-generated speech and naturalistic clinical recordings remains, and we flag this explicitly as a limitation of the speech-plus-sound section for our dataset exclusively. We would also like to highlight that other components of MedMosaic do contain real clinical acoustics and not their synthetic substitutes (see Appendix A.1).
>
> ---
>
> ## W3
>
> This is a fair critique that affects reproducibility. We will add a dedicated **reproducibility appendix** specifying:
>
> - Audio sampling rates (**16 kHz** for open-source audio data; **44.1 kHz** for synthetically generated audio data)
> - Inference-time prompt templates for each QA pair type (which were identical across models)
>
> We did not use chunking explicitly, as our longest input audio was approximately **15 minutes** — well within the context window of all audio reasoning models.
>
> ---
>
> ## Q1
>
> We thank the reviewer for this remark.
>
> ---
>
> ## Q2
>
> We acknowledge the reviewer's concerns about stylistic and circular bias, and guard against them as follows:
>
> 1. The use of **Gemini-3-Flash** for framing questions and **GPT-5.1** for judging answers helps mitigate the issue.
> 2. We will extend human validation to a larger calibration subset and explicitly report results.
> 3. Based on your recommendation, we plan to use **Spearman correlation** with human judges to quantify the bias.
>
> ---
>
> ## Q3
>
> This is an excellent suggestion. We agree that quantifying how often items are answerable from question text alone is essential for assessing benchmark integrity. Our SME validation confirmed **98.6%** of pairs received the highest **"Audio Necessity"** rating, but this was human-assessed rather than empirically measured.
>
> In view of suggestions from multiple reviewers, a **no-audio baseline experiment** — providing only the question and options to each model without any audio input — will be performed, and the accuracies reported in the camera-ready version.

---

> > ### Author Rebuttal · Reviewer_pNbf · 2026-04-05
> >
> > My concerns have all been solved

---

### Decision · Program_Chairs · 2026-04-30

**Decision:**

Accept (regular)

**Comment:**

This paper unanimously received positive reviews. Since all the reviewers show their satisfaction toward the rebuttal, it should be correctly reflected in the final version.